# A quantitative inventory of yeast P body proteins reveals principles of composition and specificity

**Wenmin Xing[1], Denise Muhlrad[2], Roy Parker[2], Michael K Rosen[1]\***

[1]Department of Biophysics, Howard Hughes Medical Institute, UT Southwestern Medical Center, Dallas, United States; [2]Department of Biochemistry, Howard Hughes Medical Institute, University of Colorado, Boulder, United States

**Abstract** P bodies are archetypal biomolecular condensates that concentrate proteins and RNA without a surrounding membrane. While dozens of P body proteins are known, the concentrations of components in the compartment have not been measured. We used live cell imaging to generate a quantitative inventory of the major proteins in yeast P bodies. Only seven proteins are highly concentrated in P bodies (5.1–15µM); the 24 others examined are appreciably lower (most $\leq$ 2.6µM). P body concentration correlates inversely with cytoplasmic exchange rate. Sequence elements driving Dcp2 concentration into P bodies are distributed across the protein and act synergistically. Our data indicate that P bodies, and probably other condensates, are compositionally simpler than suggested by proteomic analyses, with implications for specificity, reconstitution and evolution.

**\*For correspondence:**
michael.rosen@utsouthwestern.edu

**Competing interests:** The authors declare that no competing interests exist.

## Introduction

Eukaryotic cells contain numerous compartments that concentrate specific sets of molecules without a surrounding membrane (*Banani et al., 2017*; *Shin and Brangwynne, 2017*). These structures, referred to as biomolecular condensates, are related to a variety of biological processes. Examples include cytoplasmic processing bodies (P bodies) associated with RNA metabolism (*Decker and Parker, 2012*); promyelocytic leukemia nuclear bodies (PML NBs) involved in transcription, DNA damage repair, and anti-viral responses (*Lallemand-Breitenbach and de Thé, 2010*); signaling clusters in T cell activation (*Su et al., 2016*); and HP1 clusters in heterochromatin organization (*Larson et al., 2017*; *Strom et al., 2017*). Many condensates form through self-assembly of multivalent molecules, including proteins composed of folded domains and/or disordered regions, RNA and DNA, and chromatin (*Li et al., 2012*; *Kato et al., 2012*; *Nott et al., 2015*; *Su et al., 2016*; *Banani et al., 2017*; *Gibson et al., 2019*). This process can lead to both liquid-like and solid-like structures (*Banani et al., 2017*; *Shin and Brangwynne, 2017*; *Alberti and Dormann, 2019*). Some condensates have been shown to respond sharply to changes in concentration of key components or regulators, salt and/or temperature, suggesting that they form through highly cooperative assembly mechanisms, including phase transitions (*Beutel et al., 2019*; *Brangwynne et al., 2009*; *Falahati and Wieschaus, 2017*; *Li et al., 2012*; *Riback et al., 2017*; *Saha et al., 2016*; *Smith et al., 2016*; *Wang et al., 2014*; *Weber and Brangwynne, 2015*). The activities of biomolecular condensates are thought to derive from the assembly of specific collections of functionally related molecules into a unique physical environment. Thus, understanding condensates as chemical entities requires knowledge and understanding of their compositions.

Large-scale proteomics studies have been performed to determine comprehensive inventories of molecules that localize to different condensates, including stress granules (*Jain et al., 2016*; *Markmiller et al., 2018*; *Youn et al., 2018*), P bodies (*Hubstenberger et al., 2017*; *Youn et al.,*

2018), and nucleoli (*Andersen et al., 2002*). There are also extensive studies on the localization of individual molecules to different condensates. Cumulatively, these studies suggest that condensates are biochemically complex, containing tens to hundreds of types of proteins and RNAs that show dense and complex patterns of molecular interactions. Although some transcriptomic studies and a few analyses of individual proteins have been quantitative (*Khong et al., 2017*; *Klingauf et al., 2006*; *Leung et al., 2006*; *Wheeler et al., 2017*), proteomic studies to date have been only qualitative. Therefore, both the relative and absolute concentrations of proteins in condensates, and their relationships to molecular connectivity, are largely unknown. Additionally, the relationships between concentration and dynamics of molecules in condensates have not been systematically explored. Moreover, while some components are shared between different condensates, many are uniquely concentrated in specific condensates. It is still unclear how the substantial specificity is determined in vivo.

To address these issues, we sought to examine the composition of a complex cellular condensate in a quantitative manner. We performed a systematic analysis of yeast P bodies, an archetypical biomolecular condensate. P bodies are protein- and mRNA-rich cytoplasmic condensates conserved from yeast to mammals (*Ingelfinger et al., 2002*; *Sheth and Parker, 2003*; *van Dijk et al., 2002*). They are thought to participate in RNA metabolism, modulating mRNA decay and acting as sites of RNA storage during cellular stress (*Aizer et al., 2014*; *Bhattacharyya et al., 2006*; *Cougot et al., 2004*; *Sheth and Parker, 2003*). Taking advantage of an available yeast GFP library, we used quantitative fluorescence microscopy to measure the absolute concentrations of 31 P body resident proteins within the condensates and in the surrounding cytoplasm. We also measured their dynamic properties using fluorescence recovery after photobleaching (FRAP). We find that P body proteins segregate into two groups based on their concentrations in P bodies and dynamics. Members of the first group (Dcp2, Edc3, Pat1, Xrn1, Lsm1, Dhh1, Upf1) are highly concentrated in P bodies (5.1–15 μM) and exchange slowly with the cytoplasm. Except for Xrn1, these all have high connectivity, interacting with multiple P body components. Partitioning of several of these highly concentrated (HC) proteins (Dcp2, Edc3, Pat1, and Xrn1) into P bodies is correlated, suggesting that they assemble cooperatively, consistent with their high connectivity. All proteins known to contribute strongly to P body formation through genetic studies are in this HC group. In contrast, members of the second less concentrated (LC) group are at substantially lower concentrations in P bodies (all except Sbp1, 0.7–2.6 μM) and exhibit faster dynamics. These all have low connectivity. In a molecular dissection of Dcp2, we find that the N-terminal domain, multivalent C-terminal domain, and central high affinity Edc3 binding site all contribute to partitioning of the protein into P bodies and to its dynamic exchange with the cytoplasm. Moreover, the N-terminal and C-terminal domains can act synergistically to promote recruitment of Dcp2 into P bodies. This highly distributive organization suggests a ready means of quantitatively modulating condensate composition during evolution. These data suggest that while a condensate may contain many components, only a small number are highly concentrated there.

## Results

### Strategies for quantification of P body proteins

In order to understand the chemical nature of a condensate it is important to both know its members, and also their concentrations. Two criteria must be met to accurately measure protein concentrations in a condensate based on fluorescence intensities alone: 1) the size of the condensate must be larger than the point spread function (PSF, i.e. the diffraction limit) of the microscope used in the analysis since the fluorescence intensities are diluted for small objects (*Fink et al., 1998*); 2) the composition of the condensate cannot change over time, allowing reliable comparisons among multiple condensates in multiple cells. P bodies in wild type *S. cerevisiae* are typically diffraction-limited in size under normal conditions. But they become larger when mRNA decay is decreased, for example when mRNA decay proteins such as Dcp1 are deleted or when cells respond to stresses such as glucose starvation (*Teixeira et al., 2005*; *Teixeira and Parker, 2007*). Initially we analyzed GFP-tagged proteins expressed in a *dcp1Δ* strain, because the effects of stress are time-dependent and thus more difficult to perform systematically measurements across a series of strains. We individually expressed 31 reported P body resident proteins tagged with GFP at their C-termini in their

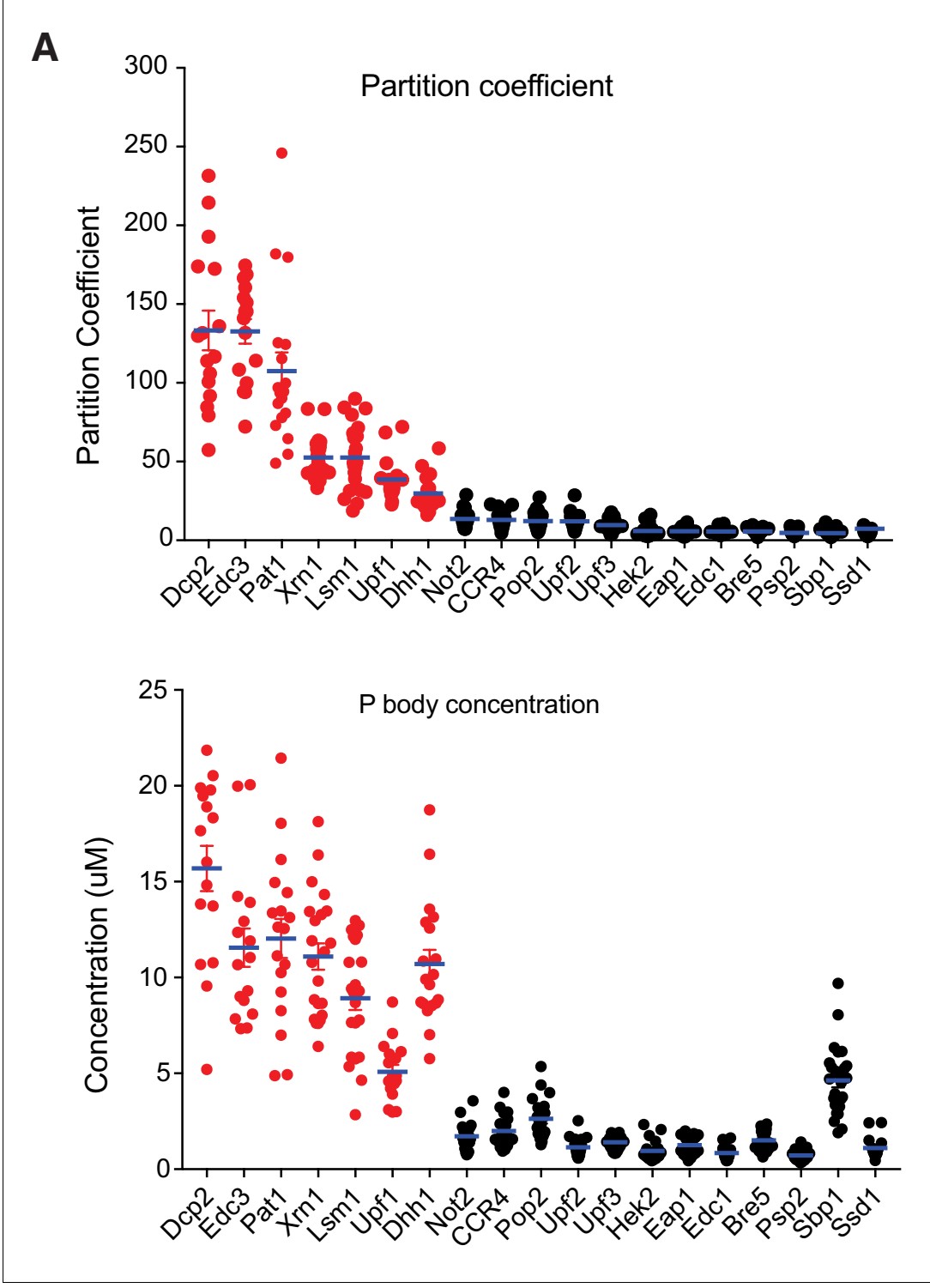

**Figure 1.** Seven proteins are highly concentrated in P bodies. (**A**) Partition coefficients (PCs) of 19 P body proteins. Plots show PCs and mean values (blue lines)± standard error of the mean (SEM). (**B**) Absolute concentrations in P bodies of 19 P body proteins. Plots show absolute concentrations in P bodies and mean values (blue lines)± SEM. For (**A**) and (**B**) each dot represents an individual P body. Red, HC P body proteins. Black, LC P body proteins. One P body per cell was analyzed from 16 to 25 cells for each protein.

The online version of this article includes the following figure supplement(s) for figure 1:

**Figure supplement 1.** Punctate localization of GFP tagged proteins and co-localization with P body marker.

*Figure 1 continued on next page*

*Figure 1 continued*
**Figure supplement 2.** Quantifications of protein concentrations using fluorescence intensities.
**Figure supplement 3.** Verification of quantitative measurements.

chromosomal location under their endogenous promoters (*Figure 1—figure supplement 1A*, and *Supplementary file 1*). We presume these puncta are P bodies because previous reports that the proteins localize to P bodies and in all cases tested we observed co-localization of the proteins with an mCherry-tagged P body marker, Edc3 (*Figure 1—figure supplement 1B*).

Using confocal microscopy, we limited our analyses to P bodies that were larger than the x-y PSF of our microscope (see Materials and methods), and corrected the diluting effect of the larger z-PSF based on an assumption that the structures were spherical. For each protein, we measured the absolute concentration in the P body and in the surrounding cytoplasm, based on cellular fluorescence intensity and independent calibration of fluorescence versus GFP concentration on our microscope (*Figure 1—figure supplement 2*)(see details in Materials and methods). We calculated the partition coefficient (PC) as the ratio of these two values.

We also used fluorescence recovery after photobleaching (FRAP) to measure the exchange between P bodies and the cytoplasm. Of the 31 proteins, 19 showed sufficiently punctate distributions in mid-log phase to permit analysis (PC >~2). We refer to these 19 proteins as regular P body proteins hereafter. The remaining 12 proteins were distributed relatively uniformly in the cytoplasm and their P body concentrations could not be analyzed; these may be stress or strain specific proteins that do not concentrate in P bodies under our experimental conditions (*Supplementary file 1*).

Two observations suggest that the GFP tag probably does not strongly affect protein behaviors. First, the PC values and dynamics of Edc3 and Dhh1 tagged with GFP and mCherry were similar (*Figure 1—figure supplement 3A and B*). Second, the PC values and dynamics of Dcp2 are nearly identical with an N-terminal or C-terminal GFP tag (*Figure 1—figure supplement 3C*). In a screen of this size it is not practical to validate each protein with multiple tags at multiple locations. It remains possible that some other proteins are affected by the tag.

## Seven proteins/assemblies are highly concentrated in P bodies

We obtained PC for all 19 regular P body proteins, which revealed subclasses of these molecules. Average PC values of the 19 regular P body proteins had a wide range, with a maximum of 133 (Dcp2) steadily decreasing to a minimum of ~5 (Eap1, Ssd1) (*Figure 1A* and *Supplementary file 2*). Only a few proteins had high partition coefficients, including Dcp2 (133 ± 13), Edc3 (133 ± 8) and Pat1 (107 ± 12); the majority had PC values < 20. We also obtained absolute concentrations of each protein in P bodies, which ranged from ~0.7 μM to 15 μM and revealed two notable features of P bodies.

First, seven proteins, Dcp2, Edc3, Pat1, Xrn1, Lsm1 (likely representing the entire, constitutive Lsm1-7 assembly [*Sharif and Conti, 2013*]), Dhh1 and Upf1 – had average P body concentrations of 8.9–12 μM, with a slightly higher concentration (15 μM) for Dcp2 and slightly lower concentration (5 μM) for Upf1. We refer to these components as highly concentrated (HC) P body proteins.

A second striking observation was that there was a clear, sharp distinction between highly concentrated (HC) and less concentrated (LC) proteins (*Figure 1B*). After the HC proteins most other proteins have P body concentrations < 2.6 μM, with most ~1 μM. The only exception is Sbp1 (4.6 μM), which also has a very high cytoplasmic concentration, affording it a high P body concentration even with a small PC value (*Figure 1*, and *Supplementary file 2*). Sbp1 also is not highly enriched in glucose-starved wild type P bodies (see below); for these reasons we do not include it in the HC group. Our data are consistent with previously reported measurements since the total cellular concentrations that we measured are similar to a unified quantitative protein abundance database of *S. cerevisiae* (*Ho et al., 2018*; *Figure 1—figure supplement 3D*).

Together, these data show that while yeast P bodies can contain many different proteins, only a relatively small number are highly concentrated in the compartment, and the majority of components are weakly concentrated.

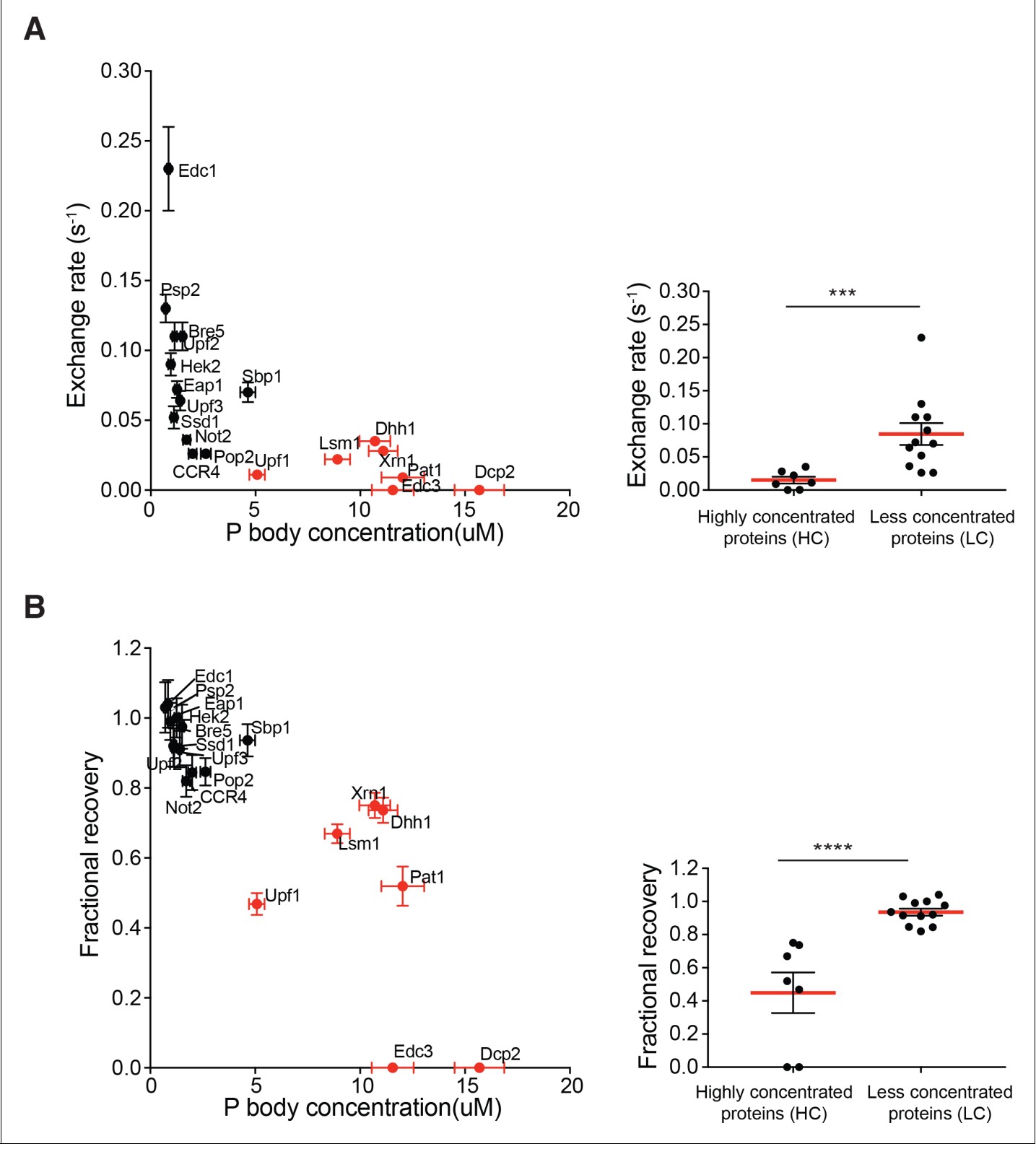

**Figure 2.** Highly concentrated proteins tend to have slow dynamics. Exchange rate (**A**) or fractional recovery (**B**) was plotted as a function of absolute concentrations in P bodies, respectively (mean ± SEM). HC and LC proteins are indicated by red and black symbols, respectively. Graphs on the right show averaged exchange rate or fractional recovery (red lines) in HC and LC groups. Analysis of significance was calculated by Wilcoxon rank-sum test. *** and **** indicate p values less than 0.001 and 0.0005, respectively.

*Figure 2 continued on next page*

*Figure 2 continued*

The online version of this article includes the following figure supplement(s) for figure 2:

**Figure supplement 1.** FRAP recovery curves of 19 P body proteins.
**Figure supplement 2.** Slow dynamics are not caused by smaller fluorescence pool in cytoplasm.
**Figure supplement 3.** Partitioning and dynamics of P body proteins in wild type strains under glucose starvation are qualitatively similar to *dcp1Δ* strains.
**Figure supplement 4.** Protein-protein and protein-RNA connections among regular P body proteins.

## Highly concentrated proteins tend to have slow dynamics

We also examined the dynamics of the 19 regular P body proteins using FRAP. For each GFP-fusion expressing strain, entire P bodies (0.4–0.8 μm) were photobleached and the fluorescence recovery curves were fit to single exponential. The exchange rate ($k$) and fractional recovery were used to assess their dynamics. This analysis revealed that the regular P body proteins exchange with the cytoplasm with very different rates and fractional recoveries (*Figure 2—figure supplement 1* and *Supplementary file 2*). At one end of the distribution, Dcp2 and Edc3 showed no measurable recovery on a 150 s timescale. Proteins such as Pat1 and Upf1, showed intermediate rates and extents of recovery ($k$ = ~0.01 s$^{-1}$, recovery = ~50%). While at the other end of the spectrum, proteins such as Sbp1 and Eap1 recovered nearly 100% in 150 s ($k$ = 0.07 s$^{-1}$).

Since we observed heterogeneity of both partitioning and dynamics, we asked whether these parameters are related. For each protein, we plotted the exchange rate ($k$), and fractional recovery against P body concentration (*Figure 2*). Although exchange rate and recovery for both the HC proteins and LC proteins spanned broad ranges, the former (Dcp2, Edc3, Pat1, Xrn1, Lsm1, Upf1, and Dhh1) tended to have slower exchange rates and less fractional recoveries than the latter (*Figure 2*). On average, for the HC group, the exchange rate was five-fold smaller than the LC group, and recovery fraction was two-fold lower (*Figure 2*).

Three observations argue that the slower recovery dynamics of the HC proteins are not simply due to smaller fluorescent pools in cytoplasm, but rather reflect different interactions in the P body. First, neither exchange rate nor fractional recovery correlates with protein concentrations in cytoplasm (*Figure 2—figure supplement 2A and B*). Second, less than 50% of the total pool of each protein is localized to the P body (<8% for most), suggesting that there are still substantial fluorescent pools in the cytoplasm (Figure 4). Third, we also performed inverse FRAP (iFRAP), which is insensitive to the size of fluorescent pools, for several proteins. In iFRAP, the entire cytoplasm except one P body was bleached and the loss of fluorescence in the P body was followed over time. In each case, the exchange rate and fractional recovery measured by iFRAP was similar to those measured by FRAP (*Figure 2—figure supplement 2C and D*).

## Partitioning of highly concentrated proteins into P bodies is correlated

We noticed that the PC values and P body concentrations of proteins varied in a broad range, for example, the PC of Dcp2 varied from 50 to 250 between different cells (*Figure 1*). In addition, the HC proteins, Dcp2:Edc3:Pat1:Lsm1:Dhh1:Xrn1, are present at roughly equimolar average levels (~10 μM), suggesting their concentrations may be correlated. To better understand the variability of partitioning, we measured the concentrations of pairs of HC proteins, differentially tagged with GFP or mCherry, in the same P bodies (*Figure 3A*). This revealed that the concentrations of paired Edc3 and Dcp2, Pat1 and Dcp2, and Xrn1 and Dcp2, are positively correlated as indicated by Pearson correlation coefficients of 0.6–0.7 (*Figures 3B, C and D*). Moreover, the ratios of the two measured protein concentrations have narrower ranges than the ratios calculated by randomizing the pairing (*Figure 3A and E*). These data indicate that partitioning of HC proteins into P bodies is correlated, suggesting they assemble cooperatively (see Discussion).

## Partitioning and dynamics of proteins are not strongly affected by Dcp1 deletion

Since the above analysis was all performed in a *dcp1Δ* strain, we wanted to measure PC and P body dynamics under a different condition to evaluate if the Dcp1 deletion made a substantial impact on

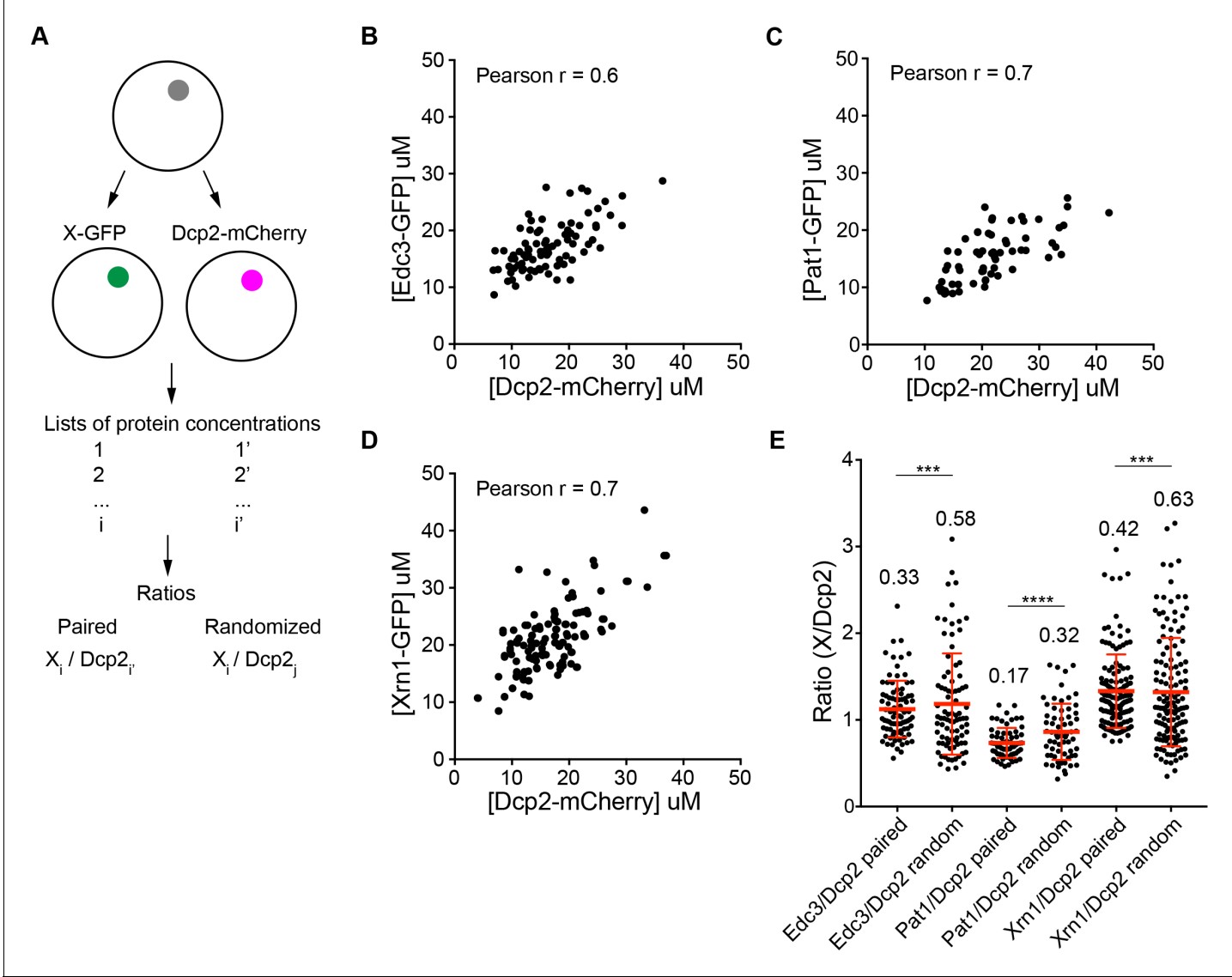

**Figure 3.** Partitioning of proteins into P bodies is correlated. (A) Schematics of calculations of ratios of paired and randomized proteins. Paired ones are ratios of X-GFP to Dcp2-mCherry in the same P bodies where i = i′. Randomized pairing are achieved by randomly generating i and j in Excel where i ≠ j. The sample size is kept the same as paired ones. (B/C/D) Concentrations of Edc3 and Dcp2 (B), Pat1 and Dcp2 (C), and Xrn1 and Dcp2 (D) are correlated in P bodies. Each dot represents one P body in a cell, 86 cells (B), 64 cells (C), and 135 cells (D) were analyzed. Pearson correlation coefficients were calculated in Prism (GraphPad ). (E) Ratios of paired proteins have tighter ranges than ratios when randomizing pairing. Plots show ratios (black dots) and mean values ± standard deviation (red lines), values are shown. Fligner-Killeen test was used to test equality of variance in R. ***, p<0.001, ****, p<0.0005.

P body composition or dynamics. For this experiment, we analyzed wild type strains after 30–60 min of glucose starvation. By several criteria, we observed similar results to our analysis in *dcp1Δ* strains.

First, similar to the *dcp1Δ* strains, during glucose deprivation, Dcp2, Edc3, Pat1, Lsm1, Xrn1, Upf1, and Dhh1, remain the most concentrated proteins in wild type P bodies, although Pat1 and Lsm1 dropped compared to the *dcp1Δ* strains (*Figure 2—figure supplement 3A and B*). Two additional proteins, Dcp1 and Pby1, also partition strongly into wild type P bodies under glucose starvation. Dcp1 binds with high affinity to Dcp2, and Pby1, in turn, binds to Dcp1, explaining its absence in *dcp1Δ* P bodies (*Krogan et al., 2006*). Only two members of the LC group, Hek2 and Sbp1, partition sufficiently to permit analysis suggesting that the partitioning of LC proteins is lower under glucose starvation.

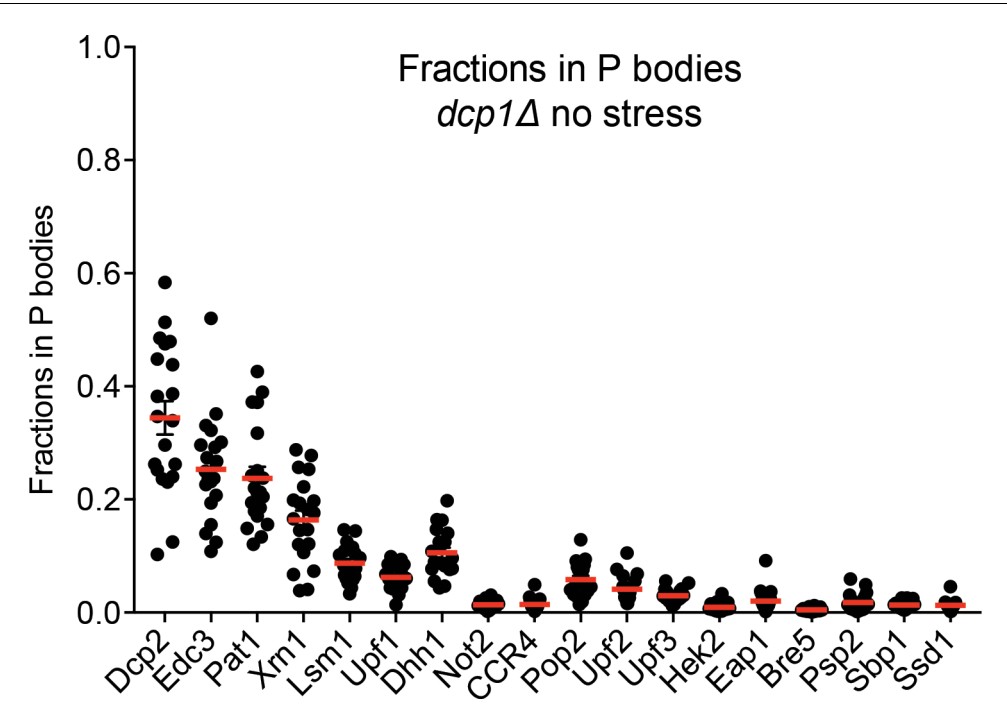

**Figure 4.** P bodies do not strongly sequester their resident proteins. Plots show fractions in visible P bodies (black dots) and mean values (red lines)± SEM in mid-log phase without any cellular stress. Each dot represents fraction of protein in visible P bodies corresponding to an individual cell. 16–25 cells were analyzed for each protein. The online version of this article includes the following figure supplement(s) for figure 4:

**Figure supplement 1.** P bodies do not strongly sequester their resident proteins.

We also observed that the dynamics of all proteins under glucose starvation were qualitatively similar to the *dcp1Δ* strains. The HC proteins have slower dynamics, although there is no clear distinction between HC and LC groups because of the limited number of the latter (*Figure 2—figure supplement 3C and D*). Nevertheless, these data suggest that partitioning and dynamics of proteins are qualitatively similar in the wild type strains under glucose starvation and the *dcp1Δ* strains, especially for the HC proteins. We thus performed all subsequent experiments in *dcp1Δ* strains, unless stated otherwise.

## P bodies do not strongly sequester their resident proteins

The biological importance of concentrating certain proteins into P bodies is unclear. One proposed function is that the P body could sequester molecules, inhibiting their activities in the cytoplasm. Similarly, P bodies have been suggested to store mRNAs or proteins under cellular stress, which could then be returned to the cytoplasm when the stress is resolved (*Aizer et al., 2014*; *Bhattacharyya et al., 2006*; *Brengues et al., 2005*). To examine the efficiency of protein sequestration, we quantified the fractions of each P body protein in the compartments ($F_P$). We first measured $F_P$ in mid-log phase without stress. Under these conditions, the most concentrated proteins, Dcp2 and Edc3, are on average ~30% sequestered in observable P bodies (*Figure 4*). For the other highly concentrated proteins (Pat1, Xrn1, Lsm1, Upf1, and Dhh1), about 10% is sequestered in visible P bodies. $F_P$ is even smaller for less concentrated proteins such that only about 5% of each is in visible P bodies.

We further asked if efficiency of P body sequestration changes under different conditions. We first measured the degree of P body sequestration in *dcp1Δ* strains after 4 hr of glucose starvation (*Teixeira and Parker, 2007*). Although $F_P$ increased for most of the proteins, visible P bodies still sequester less than 40% of each of the concentrated proteins (*Figure 4—figure supplement 1A*).

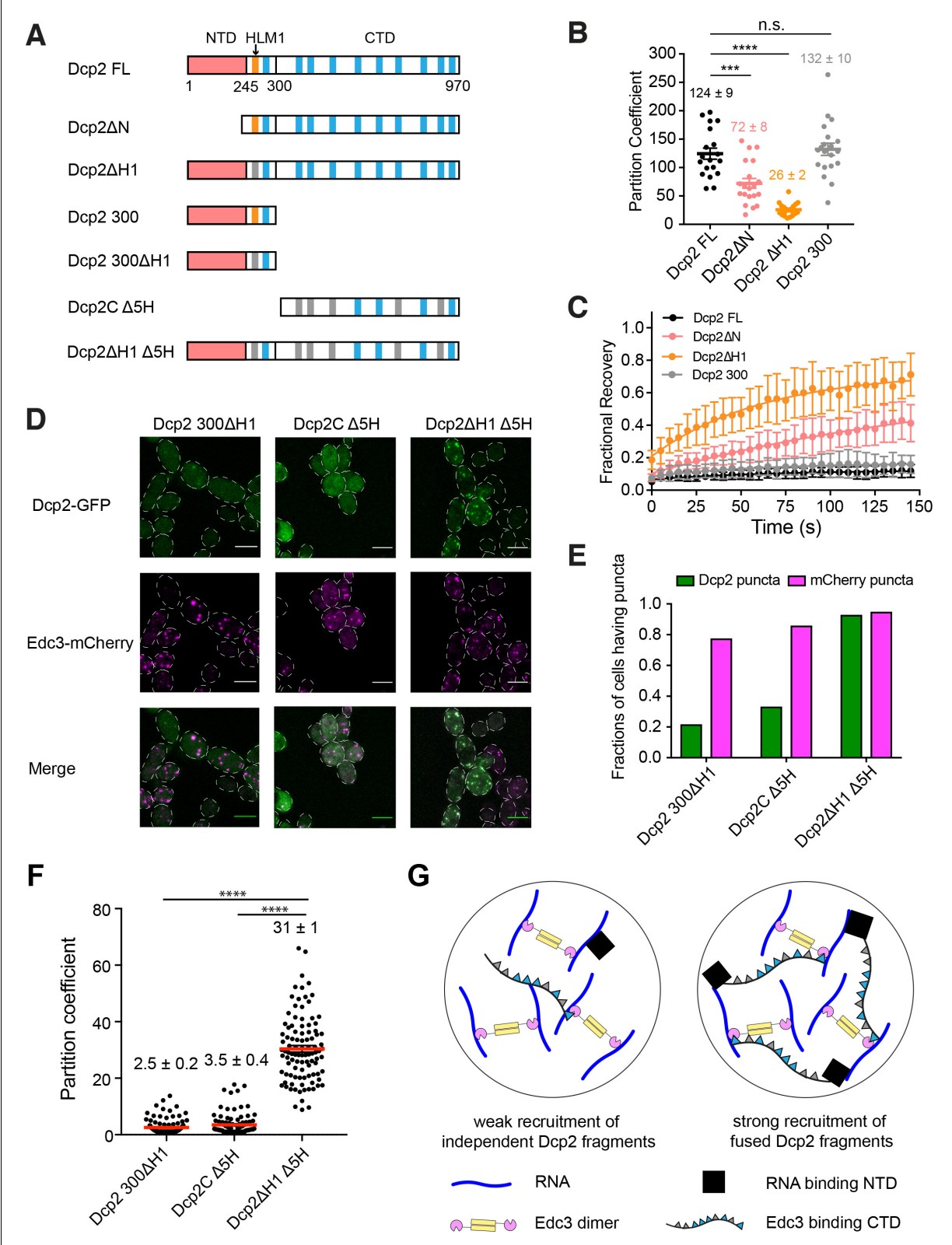

**Figure 5.** Elements controlling Dcp2 partitioning and dynamics are distributed across the protein, and contribute to specific recruitment to biomolecular condensates. (A) Schematics of domain architecture of Dcp2 FL and mutants. Red, N-terminal domain (NTD). Orange, HLM1. Blue, other 10 HLMs in C-terminal domain. Grey, inactivated HLMs. (B/C) Partition coefficients (B) and dynamics (C) of GFP tagged Dcp2 FL (black), Dcp2ΔN (red), Dcp2 ΔH1 (orange), and Dcp2 300 (grey) in *dcp1Δdcp2Δ* strain. One P body per cell was analyzed from 19 cells. PCs are shown as mean values (bold

*Figure 5 continued on next page*

Figure 5 continued

lines)± SEM (**B**). Recovery curves are shown as average of 19 P bodies ± SEM (**C**). Significance was calculated by the Wilcoxon rank-sum test. *** and **** indicate p values less than 0.001 and 0.0005, respectively. (**D**) Representative images showing *dcp1Δdcp2Δ* yeast strains expressing GFP tagged Dcp2 300 ΔH1, Dcp2C Δ5H and Dcp2ΔH1 Δ5H. Edc3-mCherry was also expressed as a P body marker. Scale bar, 5 μm. (**E**) Fractions of cells exhibiting puncta formed by each mutant. n(Dcp2 300 ΔH1)=163, n(Dcp2C Δ5H)=188, n(Dcp2ΔH1 Δ5H)=204. (**F**) Partition coefficients of GFP-tagged Dcp2 300 ΔH1, Dcp2C Δ5H and Dcp2ΔH1 Δ5H, and mean values (red lines)± SEM. 100 P bodies were analyzed. Significance was calculated by the Wilcoxon rank-sum test. **** indicates p values less than 0.0005. (**G**) Specific recruitment to P bodies can be achieved by distributing elements across Dcp2, even when they recognize distinct ligands within a condensate.

The online version of this article includes the following figure supplement(s) for figure 5:

**Figure supplement 1.** Partitioning of Dcp2 variants.

We next grew cells to stationary phase, and again, less than 50% of each protein except Dcp2 and Edc3 is in P bodies (*Figure 4—figure supplement 1B*) .

One possible explanation for the low degree of sequestration is that many P bodies are too small to be observed by standard confocal microscopy (*Rao and Parker, 2017*). This would give artifactually low $F_P$ values, as a significant fraction of P body-associated protein would not be accounted for in the integrated P body fluorescence intensity. Nevertheless, we can estimate an upper limit of $F_P$ for most proteins by assuming that all P bodies in a given cell have identical compositions independent of size, and that the $F_P$ value for the most sequestered protein (Dcp2), is, in fact one when small P bodies are properly accounted for. With this assumption, the estimated maximum $F_P$ ($F_{P,max}$) for each protein would be $F_{P,max} = F_P * 1/F_{P(Dcp2)}$. Even with this conservative estimate, most proteins are only sequestered to <20% in P bodies under all conditions.

Together, our quantifications indicate that P bodies do not strongly sequester their resident proteins under the conditions examined. Our data of course do not rule out the possibility that sequestration could be higher under different conditions, nor do they speak to sequestration/storage of RNA, which could have even higher PC values than proteins, perhaps due to non-equilibrium processes (*Ditlev et al., 2018*; *Hubstenberger et al., 2017*).

## Elements controlling Dcp2 partitioning and dynamics are distributed across the protein

We have shown that proteins concentrate into P bodies to different degrees and with distinct dynamic behaviors. We next sought to understand what molecular features could control these properties, using Dcp2, one of the most highly concentrated and least dynamic P body components, as an example. We divided Dcp2 into three distinct regions (*Figure 5A*). The N-terminal domain (NTD) of the protein possesses decapping activity and binds to Dcp1 and mRNAs (*Deshmukh et al., 2008*; *She et al., 2008*). The multivalent C-terminal domain (CTD) contains multiple short helical leucine rich motifs (HLMs) that bind to Edc3. Finally, near the center of Dcp2, the first HLM (HLM1) appears to bind Edc3 with appreciably higher affinity than all other HLMs (*Charenton et al., 2016*), and mutations to this motif impair Dcp2 partitioning into P bodies (*Harigaya et al., 2010*). We expressed different Dcp2 variants in *dcp1Δdcp2Δ* strains, in which P bodies are still formed, but they are smaller than in yeast expressing full length Dcp2 (*Figure 5—figure supplement 1A*),. The expression levels of these Dcp2 variants are similar to wild type Dcp2 as assessed by western blotting (*Figure 5—figure supplement 1B*). The analysis of these variants identified three molecular elements of Dcp2 that affect its partitioning into P bodies.

First, we found that N-terminally truncated Dcp2 (Dcp2ΔN) partitions into P bodies less efficiently than the full-length protein (Dcp2 FL), with PC values of 72 ± 8 and 124 ± 9, respectively. FRAP analysis revealed that Dcp2ΔN had an appreciably higher recovery fraction, 0.35 over 150 s, compared to Dcp2 FL, which does not recover at all in this timeframe (*Figure 5B and C*). This demonstrates that the N-terminal domain of Dcp2 promotes P body accumulation.

Second, mutating HLM1 (Dcp2ΔH1) to alanine in the full-length protein strongly decreased the PC to ~26, and increased the recovery after photobleaching to 0.65 over 150 s. Thus, both the NTD and HLM1 contribute to concentrating Dcp2 into P bodies and decreasing its exchange with the cytoplasm.

Third, additional data suggest that the other HLMs in Dcp2's C-terminal extension can contribute to P body targeting when HLM1 is missing. This is based on the observations that while truncation

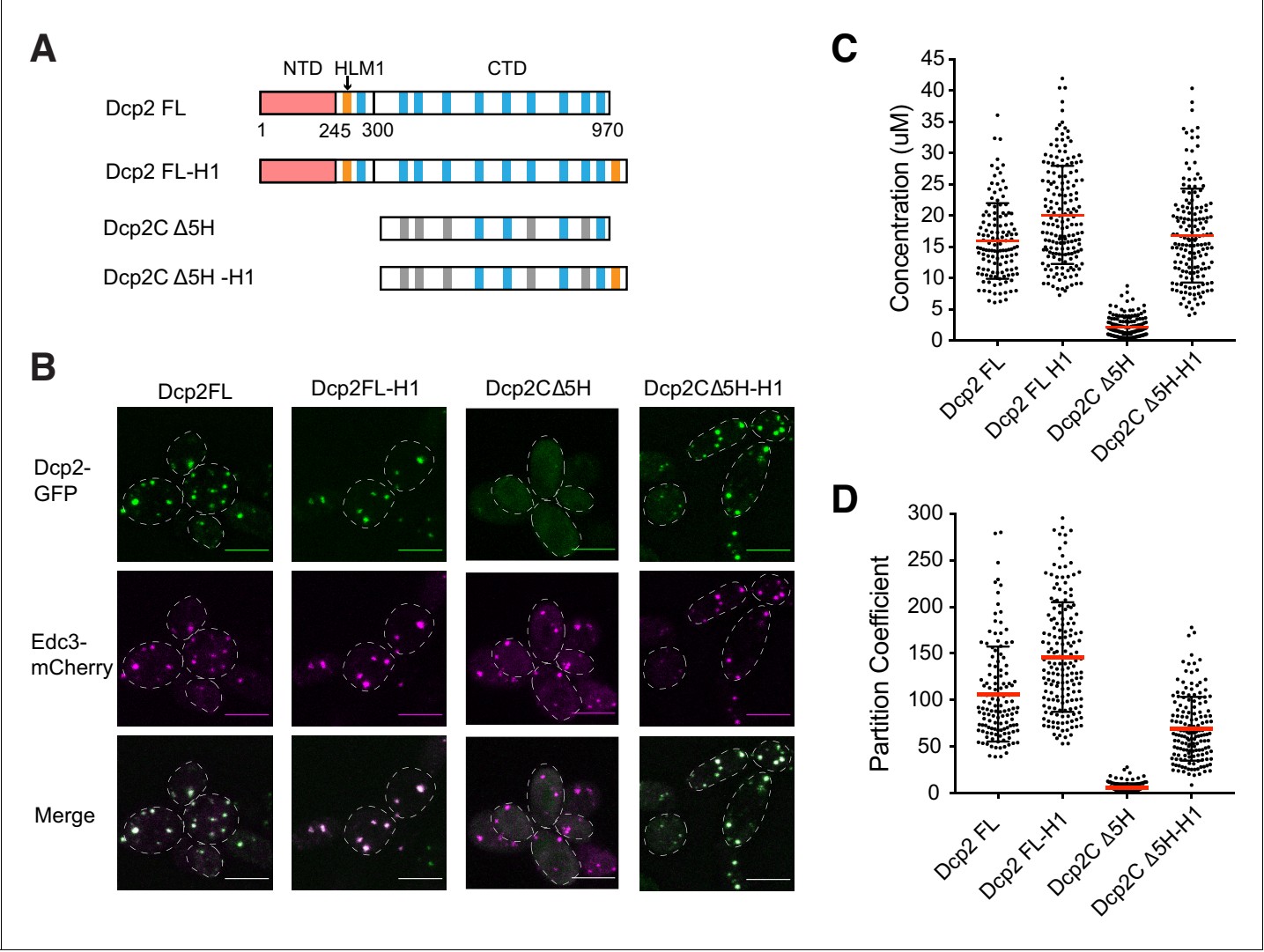

**Figure 6.** Partitioning of Dcp2 into P bodies can be saturated. (A) Schematics of domain architecture of Dcp2 FL, Dcp2 FL-H1, Dcp2C Δ5H, and Dcp2C Δ5H-H1. (B) Representative images showing *dcp1Δdcp2Δ* yeast strains expressing GFP tagged Dcp2 FL, Dcp2 FL-H1, Dcp2C Δ5H, and Dcp2C Δ5H-H1, with Edc3-mCherry co-expressed as a P body marker. Scale bar, 5 μm. (C/D) P body concentrations/PCs of Dcp2 FL (n = 49), Dcp2 FL-H1 (n = 79), Dcp2C Δ5H (n = 63), and Dcp2C Δ5H-H1 (n = 52), and mean values (red lines)± SEM.

The online version of this article includes the following figure supplement(s) for figure 6:

**Figure supplement 1.** Average cellular concentrations of Dcp2 FL, Dcp2 FL-H1, Dcp2C Δ5H, and Dcp2C Δ5H-H1 in *dcp1Δdcp2Δ* strain expressing Edc3-mCherry, and mean values (red lines)± SD.

of the CTD (Dcp2 300) had no significant effect on the PC and dynamics of Dcp2 (*Figure 5B and C*), removing the C-terminal domain from Dcp2 ΔH1 (Dcp2 300ΔH1) impaired recruitment to P bodies (*Figure 5D*).

Since the N-terminal domain and HLM1 are required for efficient partitioning and maintaining the characteristic slow dynamics of Dcp2, and the C-terminal domain can compensate when HLM1 is lacking, we conclude that elements controlling protein partitioning and dynamics are distributed across the protein.

## Recruitment elements of Dcp2 act synergistically

We next asked whether these regions can act synergistically to promote P body recruitment. We analyzed three Dcp2 mutant fragments in *dcp1Δdcp2Δ* strains: 1) Dcp2 300ΔH1, which can only interact with RNA; 2) a C-terminal Dcp2 fragment, Dcp2C Δ5H, in which five out of nine HLMs have

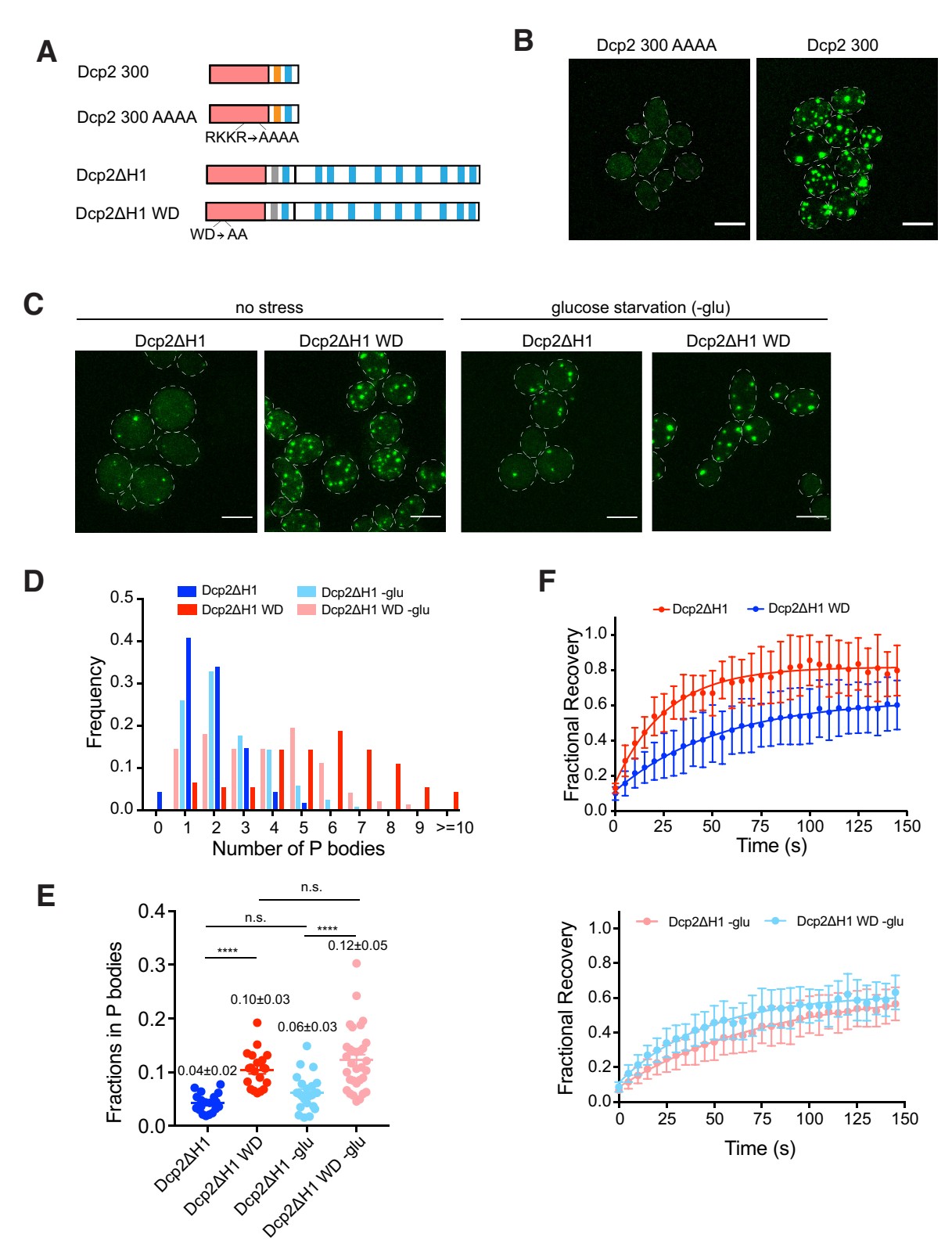

**Figure 7.** RNA binding and turnover affect Dcp2 partitioning and dynamics. (**A**) Schematics of domain architecture of Dcp2 300, Dcp2 300 AAAA, Dcp2ΔH1 and Dcp2ΔH1 WD. (**B**) Representative images of dcp1Δ*dcp2*Δ yeast strain expressing GFP tagged Dcp2 300 and Dcp2 300 AAAA. Scale bar, 5 μm. (**C**) Representative images of *dcp2*Δ yeast strain expressing GFP tagged Dcp2ΔH1 and Dcp2ΔH1 WD under normal and glucose starvation conditions. Scale bar, 5 μm. (**D**) Inhibition of RNA turnover promotes P body formation. Number of P bodies formed by Dcp2ΔH1 (blue, n = 115) and

*Figure 7 continued on next page*

*Figure 7 continued*

Dcp2ΔH1 WD (red, n = 91) under normal conditions, Dcp2ΔH1 (light blue, n = 119) and Dcp2ΔH1 WD (light red, n = 144) under 30–60 min glucose starvation. (E) Inhibition of RNA turnover promotes more Dcp2ΔH1 to partition into P bodies. Total fractions of Dcp2ΔH1(blue, n = 20) and Dcp2ΔH1 WD (red, n = 20) without stress, and Dcp2ΔH1(light blue, n = 23) and Dcp2ΔH1 WD (light red, n = 32) with glucose starvation, in P bodies. Numbers show mean values ± SEM. Significance was calculated by the Wilcoxon rank-sum test. ****, p<0.0005 (F) Inhibition of RNA turnover slows Dcp2ΔH1 exchange rate by increasing amount of RNA. Top, FRAP recovery curves of Dcp2ΔH1 (blue, n = 20) and Dcp2ΔH1 WD (red, n = 20) without stress. Bottom, FRAP recovery curves of Dcp2ΔH1 (light blue, n = 23) and Dcp2ΔH1 WD (light red, n = 32) under glucose starvation.

The online version of this article includes the following figure supplement(s) for figure 7:

**Figure supplement 1.** Mutant Dcp2 proteins express at similar levels.

been inactivated. This construct partitions into P bodies more weakly than the wild type C-terminal domain, Dcp2C, affording a larger dynamic range (*Figure 5—figure supplement 1C*); and 3) Dcp2ΔH1 Δ5H, fusion of Dcp2 300ΔH1 and Dcp2C Δ5H, which interacts with both RNA and Edc3 (*Figure 5A and G*). The three mutants expressed at similar levels in *dcp1Δdcp2Δ* strains (*Figure 5—figure supplement 1D*). We analyzed recruitment of the three proteins into P bodies using Edc3 as a P body marker.

We observed synergistic effects between these two regions of Dcp2. Specifically, Dcp2 300ΔH1 and Dcp2C Δ5H are measurably recruited into microscopic P bodies in only ~20% and~30% of cells, respectively (see Materials and methods). However, the fusion of the two fragments, Dcp2ΔH1 Δ5H, is recruited into P bodies in ~95% of cells (*Figure 5D* and *Figure 5E*). Moreover, partition coefficients of Dcp2 300ΔH1 and Dcp2C Δ5H are 2.5 and 3.5, respectively, while the partition coefficient of Dcp2ΔH1 Δ5H is 31 (*Figure 5E*). Since the product of the individual partition coefficients (8.75, which is exponentially related to the sum of the free energies of partitioning) is less than that of the fusion protein (31), we conclude that the elements act synergistically in the fusion to promote accumulation into P bodies.

Thus, while Dcp2 300ΔH1 and Dcp2C Δ5H are each recruited only weakly to P bodies, their fusion, Dcp2ΔH1 Δ5H, is recruited strongly. This indicates that recruitment elements can act synergistically when fused in cis, even when they recognize distinct ligands (in this case, Edc3 and RNA) within a condensate (*Figure 5G*). This behavior is likely mechanistically similar to avidity effects in canonical molecular interactions, where high affinity can be achieved through multivalent binding.

## Partitioning of Dcp2 into P bodies can be saturated

To further examine whether partitioning of proteins can be enhanced by adding control elements, we added HLM1 to Dcp2 FL and DcpC Δ5H, referring to these variants as Dcp2 FL-H1, and DcpC Δ5H-H1, respectively (*Figure 6A*). For the weakly partitioning DcpC Δ5H, adding HLM1 significantly increased its concentration in P bodies to ~16 μM, comparable to Dcp2 FL, and increased its partition coefficient to ~69 (*Figures 6B, C and D*), although Dcp2C Δ5H expressed higher than Dcp2C Δ5H-H1 (*Figure 6—figure supplement 1*). In contrast, adding HLM1 to Dcp2 FL, which already partitions into P bodies strongly, does not significantly further enhance either the absolute concentration or partition coefficient. Thus, binding of HLM1 to Edc3 can contribute strongly to Dcp2 partitioning. However, the interaction between the two wild type proteins appears to be effectively saturated, consistent with their strong correlation in concentrations in P-bodies (*Figure 3*), and increasing their affinity by adding a second HLM1 element to Dcp2 does not draw more of the protein into P bodies. In general, while partitioning into a condensate can be increased by strengthening weak interactions between a pair of components, the magnitude of such an effect is limited as binding nears saturation.

## RNA binding and turnover affect Dcp2 partitioning and dynamics

As the N-terminal domain of Dcp2 possesses both RNA binding and decapping activities, we further asked how these two interactions with RNA, an important scaffold of P bodies, affect partitioning and dynamics. Starting with the Dcp2 300 fragment, we mutated previously reported RNA binding residues, R170, K212, K216, and R229, to alanine, generating Dcp2 300 AAAA (*Figure 7A*; *Deshmukh et al., 2008*). Dcp2 300 AAAA does not partition into P bodies, despite

being expressed similarly to Dcp2 300 (*Figure 7—figure supplement 1A*), suggesting that RNA binding is important for Dcp2 partitioning (*Figure 7B*).

To investigate the effect of catalytic activity on Dcp2 partitioning and dynamics, we made mutations in the Dcp2ΔH1 construct, affording a better dynamic range than the wild type protein (*Figure 5B and C*). We mutated W50 and D54, which were previously implicated in RNA cap recognition and hydrolysis, to alanine, giving Dcp2ΔH1 WD (*Figure 7A*; *Charenton et al., 2016*; *Floor et al., 2010*). We then analyzed Dcp2ΔH1 and Dcp2ΔH1 WD in a *dcp2Δ* strain, where the former protein should reconstitute significant mRNA decapping activity while the latter should not.

We observed that formation of P bodies was promoted by expressing Dcp2ΔH1 WD (*Figure 7C*), which is consistent with previous observations that mutations that block decapping catalysis lead to increased P bodies (*Sheth and Parker, 2003*; *Teixeira and Parker, 2007*). Dcp2ΔH1 WD formed six P bodies per cell on average, compared to two P bodies per cell for Dcp2ΔH1 (*Figure 7D*), despite being expressed at similar levels (*Figure 7—figure supplement 1B*). Changing the catalytic rate of decapping also changed the accumulation of Dcp2 in P bodies and altered its dynamics. Specifically, ~10% of Dcp2ΔH1 WD was concentrated in visible P bodies compared with ~4% for Dcp2ΔH1 (*Figure 7E*). The exchange rate also decreased from $0.041 \text{ s}^{-1}$ to $0.017 \text{ s}^{-1}$ and fractional recovery decreased from 0.85 to 0.65 in the WD mutant (*Figure 7F*).

To further test whether these changes are due to increasing amounts of cellular mRNA or to disrupting catalytic activity of Dcp2 per se, we analyzed these two variants after 30–60 min of glucose starvation, conditions in which mRNA is increased due to translation repression. We observed that under these conditions Dcp2ΔH1 and Dcp2ΔH1 WD showed similar dynamics (exchange rates of $0.021 \text{ s}^{-1}$ and $0.013 \text{ s}^{-1}$, respectively, and fractional recoveries of 0.57 and 0.60, respectively, *Figure 7F*) and a similar distribution of the number of P bodies per cell (*Figure 7D*), suggesting that the differences observed in the non-starved conditions were due to differences in mRNA concentration. However, even in the starved conditions, the Dcp2ΔH1 WD mutant still shows a larger total fraction in P bodies than the Dcp2ΔH1 protein (*Figure 7E*), indicating that catalytic activity does control some aspects of P body formation. Together these data suggest that changes in the partitioning and dynamics of the Dcp2 WD mutant are caused by a combination of increase mRNA concentration and loss of catalytic activity.

## Discussion

This work presents the first quantitative description of an RNP granule, which is representative of the broader class of non-membrane bound organelles referred to as biomolecular condensates. This analysis has exposed features of yeast P bodies that should be generalizable to other condensates.

### Two classes of P body components

A major contribution of this work is to demonstrate two distinct classes of proteins within yeast P bodies. Members of one class, which we name the highly concentrated (HC) P body proteins, are highly enriched, with PC ≥30 and P body concentrations > 5 µM. These include Dcp2, Edc3, Pat1, the Lsm1-7 complex, Xrn1, Dhh1, and Upf1 (colored red in *Figure 1*). On average, these proteins show slower dynamics of exchange from P bodies and a small fraction of exchangeable molecules. In contrast, the less concentrated (LC) P body proteins, Not2, CCR4, Pop2, Upf2, Upf3, Hek2, Eap1, Edc1, Bre5, Psp2, Sbp1 and Ssd1, have PC <15, lower concentrations in P bodies, and show rapid exchange with the cytoplasm and a large fraction of exchangeable molecules. Thus, despite their apparent complexity from the literature, yeast P bodies are made up of a group of highly concentrated proteins, with much lower concentrations of additional factors. This illustrates a likely general principle that biological condensates that appear quite complex by qualitative analyses of their components, may in fact have a much simpler primary complexity and organization.

The differences between HC and LC P-body proteins are probably closely related to their intermolecular connectivity patterns within P bodies, which are listed in *Supplementary file 3* based on literature data. With the exception of Xrn1 (see below), the HC proteins all have high valency of interaction (number of interacting molecules) and high connectivity to other P body proteins and RNA (≥4 directly interacting molecules). This connectivity should afford increased partitioning into the condensate and reduced exchange rates and fractional recovery with the surrounding cytoplasm. It should also enhance the ability of the HC proteins to assemble, and thus contribute significantly to

P body formation. In contrast, all LC proteins have low valency and low connectivity, and usually interact with RNA and/or only one HC protein. These features should generally afford lower P body concentrations and more rapid and complete exchange. Since many of the LC components are RNA binding proteins (*Supplementary file 3* and *Figure 2—figure supplement 4*), we suggest they are recruited into P bodies by the high local concentration of RNA. The LC components with the highest PC (Pop2, Ccr4, Upf2, Upf3) all also have direct interactions with a HC P body component (Pop2 interacting with Dhh1, Upf2 interacting with Upf1), consistent with the idea that partitioning is determined by cooperative interactions of a given protein with multiple components within a condensate (see below). Thus, connectivity, and its molecular underpinning, valency of interaction, is likely to provide significant predictive power in understanding the composition of condensates.

We note that while connectivity is significantly correlated with P body concentration and dynamics, the correlation is not perfect, likely because affinity and energy-consuming processes, in addition to connectivity, can play a significant role. For example, the PC of the 1–300 fragment of Dcp2, which includes the HLM1 high affinity binding site for Edc3, is 132. In contrast, the PC of the Dcp2ΔH1 Δ5H, where HLM1 is replaced by lower affinity binding sites for Edc3 in the C terminal extension of Dcp2, is only 31. Similarly, the HC protein Xrn1 is concentrated in P bodies to ~10 μM even though it only has two connections to other P body components. This is presumably due to the high affinity of the protein for RNA (*Chang et al., 2011*; *Banani et al., 2016*). Finally, the exchange dynamics of Dhh1 appear to be governed by its rate of RNA-dependent ATP hydrolysis (*Mugler et al., 2016*). Thus, while connectivity patterns can be a useful guide to condensate composition and behavior, other molecular details must be considered to develop a complete understanding.

## Assembly of P bodies through cooperative interactions

Our data also provide insights into the nature of P body assembly. The positive correlations between HC protein concentrations (*Figure 3*) suggest that these molecules partition into P bodies cooperatively. This cooperativity likely arises from the high connectivity among the HC proteins, such that partitioning of one molecule promotes partitioning of others that interact with it directly and indirectly. Given the high concentration and multivalency of HC P body components one would predict that they would all be able to contribute to P body assembly. Indeed, this is what is seen in the literature. The three most enriched P body components, Dcp2, Pat1, and Edc3, are the three major proteins shown to affect P body assembly (*Buchan et al., 2008*; *Decker et al., 2007*; *Teixeira and Parker, 2007*). Other HC proteins, Dhh1, Upf1, and the Lsm1-7 complex can contribute to P body assembly as well (*Decker et al., 2007*; *Sheth and Parker, 2006*; *Hondele et al., 2019*; *Rao and Parker, 2017*). In contrast, none of the LC components are required for efficient P-body assembly. This illustrates a second principle that interactions that produce condensates are distributed across their highly-valent components. These act with differing degrees of cooperativity to promote formation of the larger assembly. This principle can explain why smaller P-bodies can still assemble in cells lacking one HC protein component (*Rao and Parker, 2017*).

It is notable that in the *dcp1Δ* strains, the HC components, Dcp2:Edc3:Pat1:Lsm1:Dhh1:Xrn1, are present at roughly equimolar concentrations (~10 μM), with the exception of Upf1, which is ~2 fold lower. This suggests that perhaps these components form a discrete, stoichiometrically defined RNP complex which then assembles to higher order to form the condensate. However, our data speak against this extreme of high cooperativity. First, the ratio of Dcp2 to Edc3 ranges from 0.5 to 2.3 in our analysis, inconsistent with a strictly defined ratio expected from a stoichiometric complex on every mRNA. Cast differently, the cooperativity in P body concentrations illustrated in *Figure 3* is significant, but not as high as expected from a discrete complex. Further, the P body concentrations of Pat1 and the Lsm1-7 complex drop about two fold in wild type strains under glucose starvation conditions relative to the *dcp1Δ* strains, again speaking against a discrete assembly and suggesting that the interaction network of the HC proteins could be altered under different conditions.

An unanswered question is whether intermolecular RNA-RNA interactions contribute to P body assembly. This possibility is suggested by the robust self-assembly of RNA, the roles of intermolecular RNA-RNA interactions in stress granule assembly (*Van Treeck et al., 2018*; *Tauber et al., 2020*), the observation that RNA is required for P body formation (*Teixeira et al., 2005*), and the fact that every P body protein interacts with RNA (*Supplementary file 3* and *Figure 2—figure supplement 4*). P body formation clearly requires protein interactions since Dcp2, Edc3, Dhh1, Lsm1-7 complex

and Pat1 have all been genetically shown to promote P body formation (*Decker et al., 2007*; *Teixeira and Parker, 2007*; *Rao and Parker, 2017*; *Hondele et al., 2019*). In some cases, the assembly of P bodies can be directly attributed to specific protein-protein interactions, such as the requirement for Edc3 dimerization to promote P body formation (*Ling et al., 2008*). Moreover, several P body proteins either in isolation, or in mixtures, can undergo LLPS to create P body like assemblies in vitro (*Fromm et al., 2014*; *Schütz et al., 2017*). Whether the mRNAs in P bodies simply serve as a platform to facilitate interactions between HC P body components, or also form intermolecular RNA-RNA interactions contributing to P body formation remains to be seen.

## General principles of scaffolds and clients in natural condensates

We previously proposed that the components of biomolecular condensates could be classified into two groups, scaffolds and clients (*Banani et al., 2016*; *Ditlev et al., 2018*). Scaffolds are defined as components that are required for condensate formation, while clients are not necessary for integrity of the condensate, but are recruited by interacting with scaffolds. As corollaries, the model posited that there should generally be few scaffolds and many clients in natural condensates, and that scaffolds should have high valency of interaction elements, while clients should have lower valency. In addition, scaffolds should have higher concentration within the condensate than clients, since the former recruit the latter. The model was generated based largely on the behaviors of engineered multivalent macromolecules, where the distinction between scaffold and client was stark by design. In this view, the yeast P body scaffolds include RNA (*Teixeira et al., 2005*) as well as Dcp2, Edc3, and Pat1, Lsm1-7 and Dhh1 (*Decker et al., 2007*; *Teixeira and Parker, 2007*; *Hondele et al., 2019*), since deletion of these factors impairs formation of the condensate. Notably, all of these proteins are in the HC group and have high valency of interaction and high connectivity. Under the same conditions, no molecules in the LC group are known to impair P body formation (although when cells are at high density, deletion of Sbp1 produces smaller P bodies [*Segal et al., 2006*]). Thus, information on the relative concentrations of proteins in a condensate (the HC vs LC distinction) is likely to aid prediction of molecules that play significant roles in generating the structure (and be imperfectly correlated with connectivity, as described above).

Yet a simple scaffold/client classification does not account for the differences in effect of the various deletions; for example loss of RNA completely eliminates P bodies, whereas loss of the other factors has only a partial effect. Thus, rather than a black-and-white classification, we have come to believe that scaffold and client are better used as descriptors, where condensate components are more scaffold-like or more client-like depending on the degree to which their deletion affects the cooperative assembly of other components. Scaffold-like components contribute more strongly to condensate formation; their deletion should decrease the size and number of condensates. Client-like components should have lesser or no such effects. In the context of phase transitions, the descriptor characterizes the degree by which a given condensate component influences the multidimensional phase boundary of the system, both the position of the boundary and its shape, since effects of deletion are unlikely to be evenly distributed across all molecules in a condensate (see below). Deletion of scaffold-like molecules will produce larger changes and deletion of client-like molecules will produce more subtle effects.

Intuitively, scaffold-like components should have higher connectivity among molecules in the condensate, and will be more central in the interaction network, whereas client-like molecules should have low connectivity and be more peripheral in the network. A similar concept has been described in the context of stress granules recently (*Yang et al., 2020*). In addition to affecting formation of the condensate, highly connected molecules are expected to have greater influence on the composition of the structure. Deletion of scaffold-like molecules will change partitioning of numerous components, while deletion of client-like molecules will only affect partitioning of their immediate interacting components in the network. Importantly, since connectivity in a condensate is not homogeneous (different molecules have different connectivity), deleting molecules is likely to have heterogeneous effects on composition, changing the relative ratios of components. Thus, the thermodynamics of forming a condensate and the composition of the resulting structure will be coupled, based on the connectivity patterns in the interaction network. Further exploration of these ideas will require large-scale, systematic analysis of condensate composition in the presence of individual deletions of multiple components.

## Compositional specificity

A major question regarding biomolecular condensates is how the composition of the compartments is determined in vivo. Our data suggest that recruitment of Dcp2 into P bodies is distributed across a large number of interaction elements, including a high affinity Edc3-binding motif, an RNA binding domain and weak multivalent interactions with Edc3. This organization provides several insights into compositional specificity that should be generalizable to other molecules.

First, simple mass action will concentrate components that bind with high affinity to scaffolds. For example, adding the high affinity HLM1 to the weakly partitioning Dcp2C Δ5H promotes its partitioning into P bodies to a level similar to that of wild type Dcp2. High affinity binding to RNA likely also explains the strong partitioning of Xrn1 into P bodies, even though it makes few known interactions with other components and its loss does not have deleterious effects to formation of the condensates (i.e. it is more client-like than scaffold-like in its properties).

Second, increasing affinity for a condensate component does not increase partitioning if affinity is already high. We found that adding a second HLM1 element to full length Dcp2, which should increase affinity for Edc3, does not increase its partition coefficient. The upper bound may be determined by limiting concentrations of scaffold-like interaction partners. This also indicates that partitioning may be more readily tuned when it is mediated by weak multivalent interactions rather than by high affinity interactions.

Third, protein elements that bind scaffold-like components weakly will not partition strongly into a condensate, but when two such elements are fused together, even if they bind different scaffold-like components, they can be recruited strongly. We found that two fragments of Dcp2, which bind RNA and Edc3, respectively, partition only weakly into P bodies individually. Yet when fused together, they are recruited strongly. This behavior is analogous to previous observations that cooperativity between IDRs and folded domains can promote recruitment into both phase separated droplets in vitro and P bodies in cells (*Lin et al., 2015*; *Protter et al., 2018*). Such effects, similar to avidity effects in canonical molecular interactions, will greatly narrow the specificity of recruitment, even for scaffolds that individually bind ligands promiscuously. This mechanism also provides ready routes for evolution of new clients through genetic rearrangements that fuse together multiple low-affinity interaction elements. Distributing recruitment across a large number of interaction elements would also render composition less susceptible to mutations, which could lead to evolutionary selection. A similar mechanism may be applied to RNA partitioning as well, since a predominant metric for mRNA partitioning into P bodies or stress granules is length, which may be a simple proxy for the number for interactions (*Khong et al., 2017*; *Matheny et al., 2019*).

## Quantitative considerations of condensate function

Quantitative analyses as we have performed play an important role in assessing the functions of condensates. For example, several condensates, including P bodies, have been proposed as sites for sequestration/storage of biomolecules, in part because they appear as qualitatively bright puncta by microscopy, and in part because their disruption can activate certain processes (*Arimoto et al., 2008*; *Decker and Parker, 2012*; *Li et al., 2000*). However, when total condensate volume (typically <1–2% of cytoplasm/nucleoplasm), and partition coefficients (2 ~ 200) are quantified, it is evident that only a small fraction of most molecular species are sequestered within condensates. (*Figure 4*, *Leung et al., 2006*; *Rao and Parker, 2017*; *Wheeler et al., 2017*). While in some processes small changes in the amount of available species in the cytoplasm/nucleoplasm could have functional consequences, in others, different mechanisms must be considered. For example, rather than sequestration, an inhibitory catalytic function within a condensate could also explain the activation of a process upon condensate disruption. Such considerations will further advance the already significant impact of biochemical reconstitutions of phase separation on understanding the functions of biomolecular condensates (*Fromm et al., 2014*; *Schütz et al., 2017*; *Woodruff et al., 2017*).

## Conclusions

Together, our work suggests that condensates may generally be organized around a relatively small number of highly concentrated, less dynamic scaffold-like components. This construction would provide a relatively simple route for a condensate to appear during evolution, in that only a small number of proteins would need to develop the ability to assemble cooperatively. Once this structure

was established, other proteins could evolve the ability to interact with the scaffold-like components and consequently be recruited into the compartment to different degrees determined by their connectivity, interaction affinity and cooperativity among different interaction elements. A composition of this nature also indicates that condensates are compositionally less complicated than suggested by proteomics studies, where tens to hundreds of proteins have been annotated as residents of particular condensates. Our quantitation indicates that most components are present in only small amounts, and the majority of the protein mass derives from only a few types of molecules. Such an understanding greatly simplifies efforts to reconstitute condensates in vitro, and can frame models of their biochemical functions.

## Materials and methods

### Yeast strains
Yeast strains used in this study are listed in *Supplementary file 1*. GFP or mCherry tagged proteins used to generate *Figures 1–4* are expressed from their endogenous locus. Yeast strains carrying plasmids were constructed using lithium acetate-based transformation (*Gietz and Schiestl, 2007*).

### Plasmid construction
Plasmids used in this study are listed in *Supplementary file 4*. All Dcp2 mutants were expressed under the *DCP2* promoter on a low-copy centromeric plasmid (pRP1902) as previously reported (*Harigaya et al., 2010*). Dcp2 point mutations were made by site-directed mutagenesis using KOD Xtrem Hot start DNA Polymerase followed by Dpn1 digestion. Dcp2C Δ5H, and Dcp2ΔH1 Δ5H were constructed by Gibson assembly into the vector used for Dcp2 FL, pRP1903 (*Harigaya et al., 2010*). N-Dcp2 was also constructed from pRP1903.

Yeast growth conditions *dcp1Δdcp2Δ* strains and *dcp2Δ* strains expressing GFP tagged Dcp2 mutants were grown in synthetic medium lacking uracil but containing 2% glucose. *dcp1Δdcp2Δ* strains expressing both GFP tagged Dcp2 mutants and Edc3-mCherry were grown in the same media also lacking lysine. Glucose starvation in *Figure 2—figure supplement 3*, *Figure 4—figure supplement 1A* and *Figure 7C* was performed by exchanging with the corresponding synthetic medium lacking 2% glucose for durations indicated in the text. Stationary stage in *Figure 4—figure supplement 1B* was achieved by growing cells for 5 days and $OD_{600}$ >6. For imaging, cells were grown at 30°C until $OD_{600}$ = 0.4 ~ 0.6, and then immobilized on concanavalin-A (Sigma-Aldrich) coated glass bottom dishes (MatTek).

### Image acquisition and analysis
All images were acquired using a Leica SP8 Laser Scanning Confocal Microscope using a 100 × 1.4 NA oil immersion objective. Images were analyzed using Fiji.

### Identification of P bodies
For strains expressing only GFP tagged protein, P bodies were identified by thresholding the GFP fluorescence intensity using the MaxEntropy algorithm in Fiji. For strains expressing both GFP tagged proteins and Edc3-mCherry *Figure 5* and *Figure 6*, Edc3-mCherry signals were thresholded (MaxEntropy) to identify P bodies, and created masks. Absolute concentrations of each Dcp2 mutants in P bodies and partition coefficients (P bodies within masks created by Edc3-mCherry signals) were analyzed as described below, using Edc3-mCherry signals to quantify size of P bodies. To determine fractions of cells having GFP puncta in *Figure 5E*, PC greater than two was chosen arbitrarily that punctate localization could be observed in GFP channel. Because the formation of and partitioning into biomolecular condensates are sensitive to protein expression levels, we eliminated cells with low expression (bottom 10% of the populations) and high expression (top 10% in the populations) of GFP and mCherry.

### Measurements of absolute concentrations in P bodies and cytoplasm, and partition coefficients
To quantify the absolute concentrations of GFP-tagged proteins in P bodies, we assume that P bodies are spherical (based on the similar diameters in x and y when measured), and correct their

measured intensities based on the point spread function (PSF) of our microscope (*Fink et al., 1998*). We determined the PSF using 0.2 µm fluorescent microspheres (Invitrogen) imaged with the same optics, filters, zoom settings and pinhole settings used throughout our study. We then modeled (Matlab) the intensity-diluting effect of the PSF when imaging spheres of different sizes through convoluting the PSF with the sphere. This yielded a correction curve relating sphere diameter to the fraction of true maximum intensity actually measured in the image (*Figure 1—figure supplement 2*; *Fink et al., 1998*), assuming all fluorescence intensity derived from the sphere and none derived from the surroundings. We limited our cellular analyses to P bodies with measured diameter >0.33 µm (most are 0.4–0.8 µm in *dcp1Δ* strains), which is 1.1 times larger than the x-y PSF, and thus the size of P body can be accurately measured as the full width at half maximum intensity (FWHM) of the object. From the measured diameter in the x-y dimension, an assumption of spherical shape, and the correction curve, we determined the calibration factor (CF) for the P body intensity.

To determine P body intensity ($I_{Pbody,measured}$), we first measured the maximum intensity of P body and then drew a one pixel circle around it to find the surrounding pixels. Nine pixels including the maximal one were averaged to get the $I_{Pbody,measured}$. Since the correction was based on the assumption that all fluorescence intensity derived from the object, we applied it only to the incremental intensity of the P body over the cytoplasm ($I_{Pbody,measured}$-$I_{cyto}$); cytoplasmic intensity also contributes to $I_{Pbody,\ measured}$, but is homogenous across the cell and should not be corrected for the PSF effect. The real maximum intensity of P body($I_{Pbody}$) was thus calculated as [($I_{Pbody,measured}$-$I_{cyto}$)/ CF + $I_{cyto}$]. Cytoplasm intensity was calculated by averaging the mean intensities of three ROIs the same size as P bodies in the cytoplasm.

We used standard curves of the fluorescence intensities of GFP solutions imaged with identical parameters as yeast to convert $I_{Pbody}$ and $I_{cyto}$ to absolute concentrations in P bodies ($C_{Pbody}$) and the cytoplasm ($C_{cyto}$). Because the intensities of P bodies marked by different proteins have a large dynamic range, to avoid saturation of our camera, we imaged them using different laser powers and gain settings, and generated different GFP standard curves accordingly. Partition coefficients were calculated as $C_{Pbody}/C_{cyto}$.

## Total fractions in P bodies, average cellular concentration and number of molecules per cell

We collected z-stacks of yeast cells with a 0.22 µm step size. To calculate the number of molecules in cytoplasm ($N_{cyto}$), we first measured the cell volume. Diameters in x, y and z directions were measured manually with Otsu thresholding to determine the cell boundaries in Fiji. Cell volume was calculated as $V_{cell} = 4/3*\pi*(x/2)*(y/x)*(z/2)$, assuming that yeast cells are ellipsoidal. Previous studies have shown that the cytoplasmic volume of a yeast cell is about 67% of the total cell volume (*Uchida et al., 2011*; *Yamaguchi et al., 2011*). To calculate the number of molecules in P bodies ($N_{Pbody}$), we measured the x-y diameter of each P body in the cell and calculated its volume by assuming a spherical shape. The concentrations in the cytoplasm ($C_{cyto}$) and P body ($C_{Pbody}$) were determined as described above. $N_{cyto}$ and $N_{Pbody}$ were calculated as $N_{cyto} = V_{cell} \times 0.67 \times C_{cyto} \times N_A$ ($6.02 \times 10^{23}$). $N_{Pbody} = \left( \sum V_{Pbody} \times C_{Pbody} \right) \times N_A$ ($6.02 \times 10^{23}$). Average cellular concentration = $\left( \left( \sum V_{Pbody} \times C_{Pbody} \right) + V_{cell} \times 0.67 \times C_{cyto} \right)/V_{cell} \times 0.67$. The fraction of molecules in P bodies is thus $N_{Pbody}/(N_{cyto} + N_{Pbody})$.

## Fluorescence recovery after photobleaching (FRAP)

Selected P bodies were bleached using an 0.5 W 488 nm laser at 60% laser power. Images were collected from a single plane using a 2.5 airy unit pinhole at 5 s intervals for 150 s. Fluorescence intensities were analyzed manually in Fiji. Background intensities ($I_{background}$) were first subtracted. Because yeast cells are small, the cytoplasm may be bleached slightly while bleaching the P bodies. We thus measured the average fluorescence intensities of cytoplasm (excluding the bleached P body) before bleaching ($I_{cytobefore}$) and in the first frame after bleaching ($I_{cytoafter}$) to account for this effect. An unbleached P body was used to correct for the photo-bleaching during image acquisition in the recovery phase ($I_{unbleached}$). The corrected recovered intensities ($I_{recovery}$) were normalized to the intensities pre-bleaching ($I_{pre-bleaching}$).

$$I_{recovery} = \frac{(I - I_{background})}{(I_{unbleached} - I_{background})} \times \frac{(I_{cytoafter} - I_{background})}{(I_{cytobefore} - I_{background})}$$

$$I_t = \frac{I_{recovery}}{(I_{pre-bleaching} - I_{background})}$$

Normalized intensities were fitted to a single exponential recovery (one-phase association function in Prism) (GraphPad Software).

$$I_{(t)} = I_\infty + (I_0 - I_\infty)e^{-kt}$$

where $I_0$, $I_\infty$ and $k$ were fit as intensity immediately after bleach, intensity at long times and the rate constant for recovery, respectively. The fractional recovery was calculated as:

$$F = \frac{I_\infty - I_0}{1 - I_0}$$

## inverse FRAP (iFRAP)

For iFRAP, the whole cytoplasm except one P body was bleached three times for a total of 1.5 s using a 0.5 W 488 nm laser at 100% laser power. Fluorescence intensities were analyzed as in the FRAP experiments above. Because intensities of the unbleached P body were likely affected during bleaching, we normalized the intensity to the intensities of P body in the first frame after photobleaching ($I_{pre}$).

$$I_t = \frac{(I - I_{background})}{(I_{unbleached} - I_{background})} \times \frac{1}{(I_{pre} - I_{background})}$$

Normalized intensities were fitted to a single exponential decay (one-phase decay function in Prism) (GraphPad Software).

$$I_{(t)} = I_\infty + (I_0 - I_\infty)e^{-kt}$$

where $I_0$, $I_\infty$ and $k$ were fitted parameters. The fractional decay was calculated as:

$$F = I_0 - I_\infty$$

## Construction of connection map and calculation of centrality

Protein-protein and protein-RNA interactions were summarized from literatures as shown in *Supplementary file 3*. When counting number of interactions each protein makes to others, Lsm1-7 complex was treated as a whole entity, and all the other proteins were treated as individual molecules. The connection network was generated using Cytoscape (*Shannon et al., 2003*). Eigenvector centrality of each node was calculated using CytoNCA in Cytoscape (*Tang et al., 2015*), method in which connections to highly connected nodes contribute more to score of the questioned node than equal number of connections to less connected nodes. The size of nodes and the distance of other nodes to RNA node were manually adjusted to reflect eigenvector centrality.

## Western blot

Yeast total extracts were prepared as previously described (*Knop et al., 1999*). $1.5 \times 10^8$ cells from $OD_{600} = 0.4$–0.6 cultures were resuspended in 1150 µl lysis buffer (0.24 M NaOH, 1% β-mercaptoethanol, 1 mM EDTA, 1 mM PMSF, 5 µM Pepstatin A, 10 µM Leupeptin). After incubation on ice for 20 min, 150 µl 55% trichloroacetic acid (TCA) was added to precipitate proteins on ice for 20 min. The mixture was centrifuged at 16100 rpm at 4°C for 10 min. The pellet was resuspened in 250 µl HU buffer (8 M urea, 5% SDS, 200 mM Tris-HCl [pH 6.8], 1 mM EDTA, 5% β-mercaptoethanol, and 1% bromophenol blue) and incubated at 65°C for 10 min, followed by 16100 rpm centrifugation at RT for 5 min. The supernatant was used for subsequent analyses. Immunoblotting was performed with primary antibodies: rabbit-anti-GFP (1:2000) (Abcam), and mouse-anti-PGK1 (1:1000) (Abcam). Mouse-anti-rabbit-IgG (1:10,000) (Santa Cruz) and goat-anti-mouse-IgG (1:10,000) (Santa Cruz) were used as secondary antibodies.

## Quantifications and statistical analysis

Detailed statistics including number of cells analyzed, mean value, standard deviation and standard error of the mean are indicated in each figure legend. The Wilcoxon rank-sum test was performed using GraphPad Prism (GraphPad software). The Fligner-Killeen test was performed using R. Significance was determined as: ***, $p < 0.001$; ****, $p < 0.0005$.

## Acknowledgements

We thank Saumya Jain for discussions on performing FRAP; Salman Banani, Jon Ditlev and Allyson Rice for helpful discussions on image analysis; Simon Currie and Jon Ditlev for critical reading of the manuscript; and members of the Rosen lab for helpful discussions. This work was supported by the Howard Hughes Medical Institute (MKR and RP) and Welch Foundation (grant I-1544, MKR).

## Additional information

### Funding

| Funder | Grant reference number | Author |
| --- | --- | --- |
| Howard Hughes Medical Institute | | Wenmin Xing<br>Denise Muhlrad<br>Roy Parker<br>Michael K Rosen |
| Welch Foundation | I-1544 | Michael K Rosen |

The funders had no role in study design, data collection and interpretation, or the decision to submit the work for publication.

### Author contributions

Wenmin Xing, Conceptualization, Data curation, Formal analysis, Validation, Investigation, Methodology, Writing - original draft, Project administration, Writing - review and editing; Denise Muhlrad, Resources, Investigation; Roy Parker, Michael K Rosen, Conceptualization, Supervision, Funding acquisition, Methodology, Project administration, Writing - review and editing

### Author ORCIDs

Wenmin Xing https://orcid.org/0000-0001-6445-0615
Roy Parker http://orcid.org/0000-0002-8412-4152
Michael K Rosen https://orcid.org/0000-0002-0775-7917

### Decision letter and Author response

Decision letter https://doi.org/10.7554/eLife.56525.sa1
Author response https://doi.org/10.7554/eLife.56525.sa2

## Additional files

### Supplementary files

- Supplementary file 1. Yeast strains used in this study.
- Supplementary file 2. Dynamics and partitioning of P body proteins.
- Supplementary file 3. Protein-Protein and Protein-RNA interactions among P body proteins.
- Supplementary file 4. Plasmids used in this study.
- Transparent reporting form

### Data availability

All data have been submitted to Dryad (https://doi.org/10.5061/dryad.02v6wwq0q).

The following dataset was generated:

| Author(s) | Year | Dataset title | Dataset URL | Database and Identifier |
|---|---|---|---|---|
| Xing W, Muhlrad D, Parker R, Rosen MK | 2020 | Data from: A quantitative inventory of yeast P body proteins reveals principles of composition and specificity | https://doi.org/10.5061/dryad.02v6wwq0q | Dryad Digital Repository, 10.5061/dryad.02v6wwq0q |

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
