## [Decision Letter]

**Acceptance summary:**

The work represents an effort to analyze the enrichment and dynamic properties of approximately 20 P body proteins in yeast. The descriptive part of the study is comprehensive (31 proteins included in the initial survey, 19 whose abundance was quantified) and suggest the existence of two groups: proteins strongly enriched in P bodies – several of which are also required for P body assembly – and proteins that are less enriched – fewer of which affect P body assembly. Overall the picture is one of a dynamic condensate.

**Decision letter after peer review:**

[Editors’ note: the authors were asked to provide a plan for revisions before the editors issued a final decision. What follows is the editors’ letter requesting such plan.]

Thank you for sending your article entitled "A quantitative inventory of yeast P body proteins reveals principles of compositional specificity" for peer review at eLife. Your article is being evaluated by 3 peer reviewers, and the evaluation is being overseen by a Reviewing Editor and James Manley as the Senior Editor.

Given the list of essential revisions, including new experiments, the editors and reviewers invite you to respond within the next two weeks with an action plan and timetable for the possible completion of the additional work. We plan to share your responses with the reviewers and then issue a binding recommendation.

Summary from a reviewer: This manuscript reports on a quantitative analysis of P body composition in yeast. The authors characterize the concentration, enrichment relative to cytoplasmic pool, and dynamics of 31 reported P body proteins tagged with GFP. They identify 19 that were sufficiently concentrated in P bodies under the conditions tested and divide them into two classes: a "core group" comprising 7 proteins with high enrichment and low dynamics, and the remainder 12 proteins with low enrichment and high dynamics. The authors argue that these observations support a scaffold/client model for condensate assembly where the number of interaction partners for a given protein (valency) can be used to predict its enrichment in condensates/P bodies.

All reviewers had similar issues about the work that revolve around novelty, quantitation, the fit of the data to their model and the lack of experimental testing. If you feel that you can address these concerns expeditiously, we will consider a revised manuscript.

The main concerns of the reviewers can be summarized as follows:

1. The novelty of the work is not emphasized. Clearly it is more comprehensive than previous work of these authors, but it needs some more perspective and additional testing.

2. The authors need to consider other models for assembly than just client/scaffold. The discrepancies between the data and their model are not resolved. There are several experiments detailed that could do more to enlighten the murky conclusions (see below). For instance, rather than tagging with a single fluorophore, can they estimate composition and variation better with a two fluorophore pairwise analysis.

3. Can they provide any additional data or insight to shed light on the discrepancy that they point out between their model and the results for DCP2?

[Editors’ note: the authors submitted for reconsideration following the decision after peer review. What follows is the decision letter after the first round of review.]

Thank you for submitting your work entitled "A quantitative inventory of yeast P body proteins reveals principles of compositional specificity" for consideration by *eLife*. Your article has been reviewed by three peer reviewers, one of whom is a member of our Board of Reviewing Editors, and the evaluation has been overseen by a Senior Editor. The reviewers have opted to remain anonymous.

Our decision has been reached after consultation between the reviewers. Based on these discussions and the individual reviews below, we regret to inform you that your work will not be considered further for publication in *eLife*.

The reviewers feel that they cannot support a resubmission of this work. The reasons for this are best summarized by one of the reviewers, who states:

"I am not in favor to invite a resubmission. Unless I misunderstood something, the authors are not planning to address concerns relating to significance and the validity of their model.

The authors state that: "As the reviewers suggest, we believe that A) RNA is a key scaffold of P bodies, B) binding affinity of proteins for each other and for RNA is important and C) valency of interactions is also important".

But offer no experiments/data to support these points.

They also state that: "We did not intend for the data in Figure 7 to represent a test or validation of a valency-based scaffold/client model, or a result stemming from the previous data in the manuscript. Rather, we intended it simply as an empirical correlation that could potentially be useful in making predictions about other condensates where quantitative imaging data are not readily obtainable".

This does not seem to match the article's title, which promises "compositional principles". Their offer to image two components are the same time will not add much beyond confirming what they have already shown?

Finally, the statement that "our data show that only a handful of the many proteins in P bodies are highly concentrated in the structure. This information is key to conceptualizing the constitution of condensates. It speaks against most prevailing models of condensate composition, which cast the compartments as intractably complex, with hundreds of components, all implicitly weighted equally in terms of functional significance. Our data suggest the compartments are much simpler and provide the first routes to reliable biomimetic reconstitution".

Biomimetic reconstitution of RNA granules has already begun, and it is already been shown that few components can be sufficient to mimic condensate structure (Feric et al., 2016, Putnam et al., 2019). The statement also seems to imply that knowledge of the concentration of each factor will help identify such key components, a concept that remains untested."

Reviewer #1:

In this study, Xing et al., quantitatively investigated the composition of yeast P body using an imaging-based method. They found that only a small number of proteins, the core components, are highly enriched in P bodies. The authors demonstrated that the protein concentration within the P body is inversely correlated with the protein cytoplasmic exchange rate. Then the authors further characterized the elements that determine the concentration of Dcp2p in P bodies. By calibrating their system to measure absolute protein concentration in cells, the authors provide a method that could be used to study the composition of other types of cellular granules. The manuscript aim and structure are clear and straightforward. While many of the observations about granule dynamics were previously addressed by the Rosen's and other groups, this work represents a concerted and comprehensive attack on the near-total protein composition and their partitioning behavior in the P-body. To further improve the novelty, we suggest additional questions that could be addressed:

1) A major advance of this study is the precise concentration measurement of GFP-fusion proteins, which is done by comparing the cellular GFP intensity with standard curves. All conclusions depend on these standards and background subtraction. Even though the Rosen lab previously published a paper using a similar approach (Banani et al., 2016), the details of these calibrations and background subtraction should be reported in the supplemental materials.

2) To estimate the protein concentrations in P-bodies, the authors made a fundamental assumption: that all P-bodies have the same and constant composition. However, in Figure 1, the authors showed that the core components of P-bodies, which are also the most abundant proteins, have strong (4-5-fold) variability in concentration and partitioning coefficient. It is unclear how this variability affects the relative quantifications reported in this manuscript (Figure 3). A control experiment that could address this question would be to tag several P-bodies components with spectrally distinct fluorescent proteins (2 or 3 at the time) and verify that the expected relative ratios are recapitulated.

3) Along the same line, it is unclear whether the GFP-tagged proteins, which were used to quantify the P-body components in the *dcp1∆* background, were expressed from the endogenous locus (Table 1: YRP1936, yRP2254, yRP2237, yRP2246, yRP2230, yRP1840, yRP1736, yRP2269, yRP1844, yRP1916, yRP1842). The authors should specify this since it would have significant consequences on the quantifications if there were a mixed population of tagged and untagged proteins.

4) In Figure 2, The authors found a negative correlation between the fraction recovery rate and P-body concentration. However, the molecular mechanism behind this correlation is not discussed. The authors should expand the Discussion and link it to the interaction valency.

5) In Figures 4 and 5, the authors measured the partition coefficients of several Dcp2 mutants. It would be important to demonstrate that these mutations do not significantly affect the protein stability and expression by a western blot of these mutants.

6) In Figure 5, the authors suggest that the decapping activity of Dcp2 affects the partition coefficient and recovery rate. However, as the authors mentioned in the text, it is not clear whether this is due to an increase in cytoplasmic mRNA or a direct result from Dcp2 catalytic activity. This issue may be resolved by expressing Dcp2 variants in wild type cells and inducing P-bodies with starvation.

7) In Figure 7, the authors found a correlation between P-body partitioning and protein interaction valency. This finding would be more convincing if the authors could demonstrate that when the interaction valency of a protein increases, this protein will become more concentrated in P-bodies. For example, this can be done by fusing a client protein with a domain of a scaffold protein, like Dcp2C ∆5H. The fusion protein should have more interaction valencies, and the fusion protein would be expected to be more concentrated in the P-bodies. This experiment will also enhance the conclusion made in Figure 6.

Reviewer #2:

This manuscript reports on a quantitative analysis of P body composition in yeast. The authors characterize the concentration, enrichment relative to cytoplasmic pool, and dynamics of 31 reported P body proteins tagged with GFP. They identify 19 that were sufficiently concentrated in P bodies under the conditions tested and divide them into two classes: a "core group" comprising 7 proteins with high enrichment and low dynamics, and the remainder 12 proteins with low enrichment and high dynamics. The authors argue that these observations support a scaffold/client model for condensate assembly where the number of interaction partners for a given protein (valency) can be used to predict its enrichment in condensates/P bodies.

This model was derived by the Rosen group in a previous study that examined the in vitro phase separation properties of an artificial set of proteins with varying numbers of low-affinity binding motifs (valency). The model predicts that "scaffolds" (proteins essential for condensate assembly) exhibit high enrichment and high valency, whereas "clients" exhibit low enrichment and low valency. In accordance with the model, the authors note a rough correlation between valency and enrichment among the 19 P body proteins examined (but see below). To test the model directly, they deleted binding motifs in one predicted "high-valency" P body protein (Dcp2). Surprisingly, they found no strong correlation between valency and enrichment (Many of the Dcp2 mutants do not lie on the correlations in Figure 7. It is not clear to us how to resolve this discrepancy). They also find that some known clients (proteins not essential for P body assembly) exhibit strong enrichment, contrary to the model. Despite these discrepancies, the authors continue to argue that their findings support the scaffold/client model.

It is very surprising that the authors do not consider other models that might fit their data better. One possibility is that RNA functions as the true scaffold for P bodies and that proteins are recruited to P bodies by virtue of their affinity for RNA or for other proteins that binds RNA. In this regard, it is interesting that components of multiprotein complexes are found in roughly stoichiometric amounts in P bodies, suggesting that complexes are preserved in the condensates. In this alternative model, high affinity for RNA by one protein in the complex would be sufficient to enrich the entire complex, even though none of the other subunits in the complex bind to other P body components.

The authors also claim that their analyses suggest that a relatively small number of core proteins (7) make up most of the protein content in P bodies. However, since other P body components may have yet to be discovered, this statement can only by an approximation at this time. Furthermore, even if true, the significance of this hypothesis is unclear – proteins present in low concentrations in P bodies may still have important roles.

In conclusion, while this survey documents concentrations and dynamics for a significant number of P body proteins, it does not appear to provide significant insights into the "rules" that regulate P body composition. The model put forth by the authors is not consistent with the data and other more plausible models are not considered.

Reviewer #3:

This article by the Parker and Rosen labs investigates P bodies components with the aim to provide a quantitative description. P bodies are cytoplasmic foci first described in yeast that are sites where RNA decapping and degradation can occur. P bodies are dynamic, require RNA and are sensitive to the translational status of the cell, where translational repression and stress favor P body growth. Many years of study in the Parker lab and by others has identified a number of critical components of P body assembly and provided evidence that there is not a single factor that is absolutely necessary for P body assembly and that P body assembly can be influenced by multiple and different protein-protein and/or protein-RNA interactions. In this study, the Parker and Rosen lab have used an existing set of 31 GFP tagged P body components and compared their relative abundance and physical properties in P bodies. They find that 7 proteins are highly concentrated and less mobile, they therefore suggest that these are scaffold proteins. Multiple interaction domains apparently provide structure to these components within the granule, while other components which are less enriched are more mobile.

This is a nicely executed study of limited new insight.

- Limits to novelty. It was already known that these 9 proteins are the major proteins of the P body that also play some role in P body function. These earlier studies had also already shown that several of these proteins contain multiple interaction domains through which P body components interact.

- Limits to quantitative analysis. Analysis with confocal has clear limits for spatial resolution and the quantitative measurements are based on fluorescence intensity, not single molecule measurements. Since all components are tagged with the same fluorophore (GFP) no pairwise or higher order analysis could be used to determine variation between granules, protein occupancy, correlation between proteins, to test, as example, whether all components are in a granule at same time, and with similar concentration.

- Unclear relevance of DCP2 analysis. The analysis of DCP2 seems not add much to the conclusions of the paper. Rather than showing that this protein is multivalent as has been shown by domain dissection of other “scaffold” proteins in the past, it would have been informative if the authors had provided whether the “client” proteins have indeed fewer interaction domains.

[Editors’ note: further revisions were suggested prior to acceptance, as described below.]

Thank you for submitting your article "A quantitative inventory of yeast P body proteins reveals principles of composition and specificity" for consideration by *eLife*. Your article has been reviewed by three peer reviewers, and the evaluation has been overseen by James Manley as the Senior Editor The reviewers have opted to remain anonymous.

The reviewers have discussed the reviews with one another and the Reviewing Editor has drafted this decision to help you prepare a revised submission.

Summary

This work analyzes the enrichment and dynamic properties of ~20 P body proteins in yeast. The descriptive part of the study is reasonably comprehensive (31 proteins included in the initial survey, 19 whose abundance was quantified) and suggest the existence of two groups: proteins strongly enriched in P bodies – several of which are also required for P body assembly – and proteins that are less enriched – fewer of which affect P body assembly.

The authors examine the domains in DCP2 required for P body localization and identify multiple domains that appear to function partially redundantly. They conclude that a collection of synergistic interactions drive P body assembly, a conclusion reached by other studies as well.

Revisions:

The reviewers felt the manuscript and work makes a contribution to the field, but some revisions are necessary to take into account caveats in the interpretation of the data. For instance, reviewer 2 feels that the term "cooperativity" is not appropriately supported by the data and that mRNA is not considered as a component of the P bodies. Reviewer 3 feels that the functional properties are not sufficiently addressed, particularly in view of specificity (why are some mRNAs and RBPs excluded?).

In sum, the manuscript should be modified to take into account the limitations of the study as indicated by the two reviewers, and their suggestions for revision as indicated below.

Reviewer #2:

In this largely revised manuscript, the authors report on a comprehensive quantification of component proteins of a biological condensate, the Dcp1 depleted P bodies of yeast. Through this they identified seven “core” P body proteins that are partitioned substantially, have a high concentration and slow mobility when enriched in P bodies (Figure 1 and 2). Concentration of these seven proteins in P bodies is positively correlated, consistent with previous data showing that these proteins interact biochemically (Figure 3). The reported enrichment parameters seem similar when Dcp1 depleted P bodies are compared with wild type P bodies under glucose starvation, however Dcp1 and its binding partner Pby1 are missing and RNA degradation is defective in the P body model used throughout the analysis. Thus, while at the level of their biophysical analysis the Dcp1 depleted P-bodies seem similar to P bodies under physiological conditions, functionally these “P bodies” are distinct. This reduces the relevance of this study. In Figure 4 the authors observe that core protein concentrations in P bodies are largely independent of stress conditions, however the authors only considered the protein components, while the regulated entity of stress P bodies are mRNAs and not proteins.

Next the authors dissect DCP2 P body localization in a Dcp1, Dcp2 double null mutant, which still makes some P body (although Figure S1A only shows the presence of small “P body like” structures but no quantitative comparison to control P bodies, in wild type under glucose starvation and Dcp1 depleted). In deletions studies the authors find that various domains of Dcp2 contribute to P body partitioning and show some redundancy between the H1 domain and other helical domains in the CTP. The authors conclude that elements in Dcp2 act “cooperatively”. However, their data show additivity or at most synergy but not cooperativity. In order to use the term cooperativity, the authors need to demonstrate that the initial binding event increases the binding affinity of subsequent binding events. The authors use structure-function study of DCP2 to argue that connectivity, affinity and catalytic activity are three major factors for DCP2 P body recruitment and P body size and number. However, more specific experiments would need to be performed to strengthen this argument or to generalize the finding to other P body components or other types of RNP bodies: 1) doing domain/binding site deletion study on Edc3 or other core proteins; 2) adding valency or high affinity binding site to a client-like protein to turn it into a scaffold (sufficiency). 3) Testing the model with stress granule scaffold proteins.

Overall, this study is technically well done but rather limited in its take home message or novelty. A true functional readout is missing as the experimental set up does not relate to P body function (such as the level of RNA in different granules and decapping activity). The manuscript could easily be condensed into four figures, each making one point for the characterization of a particular type of condensate: Figure 1 and 2: enrichment of certain factors in P bodies, Figures 3 and 4 relationship of concentration between components and under differing conditions, and Figures 5 and 6 structural analysis of one component, suggesting partial redundancy between the CTD helices and Figure 7 with studies addressing RNA binding.

Reviewer #3:

In this revised manuscript, Xing, et al. analyze the enrichment and dynamic properties of ~20 P body proteins in yeast. The descriptive part of the study is reasonably comprehensive (31 proteins included in the initial survey, 19 whose abundance was quantified) and suggest the existence of two groups: proteins strongly enriched in P bodies – several of which are also required for P body assembly – and proteins that are less enriched – fewer of which affect P body assembly. From this the authors suggest that P bodies are "biochemically simpler than suggested by proteomics". This is at best a conjecture and at worst a gross oversimplification since abundance is not necessarily predictive of function – especially for enzymes.

In the second part of the study, they authors examine the domains in DCP2 required for P body localization and identify multiple domains that appear to function partially redundantly. They conclude that a collection of synergistic interactions drive P body assembly, a conclusion reached by several other studies (reviewed in Lu Na and Slavoff, 2018) including in vitro analyses (Shutz et al., 2017).

Overall, we are left with a study that will be a useful reference for future P body studies, but that does not deliver on the promise of "principles of composition and specificity". The question of specificity, in particular (why don't all RNA-binding proteins localize to P bodies?) is not examined directly. The authors only perform structure-function studies on one P body protein and never address what excludes other RNA-binding proteins from P bodies.

Additional comments:

1) The authors are not consistent in the description of the CTD. They suggest first that "N-terminal domain, HLM1, and C-terminal domain are all required for efficient partitioning and maintaining the characteristic slow dynamics of Dcp2". However, later they say that "deleting the CTD has no effect on PC of full length Dcp2". The data presented suggest that the C-terminal domain is not required (300-end) but can compensate when HLM1 is absent.

2) The intact C-terminal domain is never tested; instead the authors used a CTD with 5 of the HLM domains mutated. While the authors imply that this version of the CTD has weaker association with P bodies, no data is shown, and it is unclear from the text why this version of the CTD was used.

3) For the data in Figure 7, the authors use DCP2 constructs with reduced binding to Edc3 to investigate the role of RNA binding and decapping activity in P body association. However, it is not clear why RNA binding was studied using the Dcp2 300 mutant and decapping was studied in the Dcp2ΔH1 mutant.

4) A figure comparing the P body concentration/partition coefficient and exchange rates for the different DCP2 mutants (similar to Figure 2A and B) might be a more intuitive way to compare all the different DCP2 mutants used in this study.

5) The DCP2 structure function studies identify domains in DCP2 that promote P body localization. Whether these domains are sufficient to drive a non-P body protein into P bodies is not addressed. Without such "sufficiency" experiments, whether the results for DCP2 are generalizable for other proteins remains unclear. For example, they suggest that Xrn1 is stably recruited because of its high RNA binding affinity, however, this hypothesis does not address why other high affinity RNA binding proteins are excluded from P bodies.

6) At the end of the Discussion the authors introduce the concept of avidity where multiple interactions result in strong binding. It might be useful to introduce this concept earlier.

---

## [Author Response]

[Editors’ note: what follows is the authors’ plan to address the revisions.]

The main concerns of the reviewers can be summarized as follows:1. The novelty of the work is not emphasized. Clearly it is more comprehensive than previous work of these authors, but it needs some more perspective and additional testing.

On the issue of novelty, we feel our work adds significant new knowledge about P bodies and biomolecular condensates in general. First, our work is the only report of the absolute or relative concentrations of the major proteins within any condensate. Such concentrations are essential knowledge in considering the nature and biochemical functions of P bodies (and by extension, other condensates). For example, models of condensates as storage compartments are cast in doubt by our data in Figure 3, showing that only a small fraction of most proteins are present in P bodies. In addition, models of condensates as sites of high biochemical activity must contend with the fact that most non-core components are only concentrated < 10-fold relative to the surrounding cytoplasm and that most of the molecules are still in the cytoplasm. Condensate functions thus more likely arise from the core components, or from collections of the non-core components (e.g. in a reaction cascade). These are important concepts for the field. Second, our data show that only a handful of the many proteins in P bodies are highly concentrated in the structure. This information is key to conceptualizing the constitution of condensates. It speaks against most prevailing models of condensate composition, which cast the compartments as intractably complex, with hundreds of components, all implicitly weighted equally in terms of functional significance. Our data suggest the compartments are much simpler and provide the first routes to reliable biomimetic reconstitution.

Finally, while we have more work to do to understand the correlations of Figure 7, as an empirical predictor of the degree to which full length proteins are likely to be concentrated within a condensate, they are quite valuable to the field.

2. The authors need to consider other models for assembly than just client/scaffold. The discrepancies between the data and their model are not resolved. There are several experiments detailed that could do more to enlighten the murky conclusions (see below). For instance, rather than tagging with a single fluorophore, can they estimate composition and variation better with a two fluorophore pairwise analysis.3. Can they provide any additional data or insight to shed light on the discrepancy that they point out between their model and the results for DCP2?

Concerns about the scaffold/client model appear to arise in part from the reviewers’ misperceptions about the model itself, and about our intent in presenting the valency correlations in Figure 7. In retrospect, we understand how these misperceptions arose, and believe that an extensive revision of the text, coupled with additional data, should lead to a much crisper presentation and more accurate model. In brief, we did not intend to portray a model where valency of interactions is the only parameter important in dictating condensate composition, nor where RNA is not important. Rather, as the reviewers suggest, we believe that A) RNA is a key scaffold of P bodies, B) binding affinity of proteins for each other and for RNA is important and C) valency of interactions is also important. We also mistakenly equated “core” proteins with “scaffold” proteins and “non-core” with “client” in the sentence near the start of the Discussion quoted by the second reviewer, rather than describing these as correlated but not identical, which was our intent through the rest of the Discussion. This set off a series of concerns that can be addressed in a revised introduction and discussion.

Relatedly, we did not intend for the data in Figure 7 to represent a test or validation of a valency-based scaffold/client model, or a result stemming from the previous data in the manuscript. Rather, we intended it simply as an empirical correlation that could potentially be useful in making predictions about other condensates where quantitative imaging data are not readily obtainable. This evidently did not come across correctly, and our presentation framed the model as exclusively valency-based, which was not our intent. This can be corrected in a revision. Nevertheless, we agree with the reviewers that it is important for us to understand why the Dcp2 mutants do not fall on the same correlation as the various full-length proteins and have designed an additional set of mutants to understand this better.

The second reviewer also makes an interesting point that the ~1:1 stoichiometry of many of the core proteins suggests that these may be forming a stereotypical assembly that binds RNA and then recruits other proteins, leading to a somewhat different model for composition. We have several thoughts on this issue. First, we agree that the relative stoichiometry is striking. In fact, an earlier version of the manuscript discussed this in some detail, similarly to the reviewer’s comments. However, we eventually deleted this discussion because A) the relative stoichiometry is different in the glucose-starved wild type strains, and B) the wide range of P body concentrations of each component and lack of multi-component correlations in individual yeast cells made it hard to claim a specific assembly. Nevertheless, we agree with the reviewers that our understanding of P body composition would be appreciably strengthened by acquisition of 2-color data, where we could quantify the absolute concentrations of 2 proteins simultaneously in individual cells. While it is impractical to acquire such data on all possible P body protein pairs, we will generate strains to examine key correlations within the core group and between the core and non-core proteins. These data will allow us to distinguish the different models for P body composition.

[Editors’ note: the authors submitted for reconsideration following the decision after peer review. What follows is the decision letter after the first round of review.]

The reviewers feel that they cannot support a resubmission of this work. The reasons for this are best summarized by one of the reviewers, who states:"I am not in favor to invite a resubmission. Unless I misunderstood something, the authors are not planning to address concerns relating to significance and the validity of their model.The authors state that: "As the reviewers suggest, we believe that A) RNA is a key scaffold of P bodies, B) binding affinity of proteins for each other and for RNA is important and C) valency of interactions is also important".But offer no experiments/data to support these points.

We apologize that our initial response did not state clearly enough that our intent was to heavily revise our model, better illustrate its significance, and test it further. We have done all of these things in the revision.

As a brief summary here, we no longer present the work as testing a scaffold/client model for condensate assembly. Rather, we now provide a much more nuanced view of P body assembly, emphasizing the role of high connectivity (related to, but distinct from valency) between both proteins and RNA in producing the condensate, but also discussing the importance of binding affinity and active processes. We also discuss several other models and explain how these are inconsistent with data from our labs and others, both in this paper and the literature. As suggested by the reviewers, and discussed with the editor in April, we have now acquired two-color data (examining correlations between two P body components), and data examining the role of high-affinity binding elements in driving P body recruitment, which both prove instrumental in comparing different models.

Finally, we are now are much more explicit in describing the novelty of our findings, which includes, among 5 points detailed in the accompanying document: A) the discovery that despite the large number of molecules present in P bodies, only a small number (7) are highly concentrated there with large partition coefficients and B) the finding that intermolecular connectivity plays a key role in governing the concentrations of molecules in P bodies.

They also state that: "We did not intend for the data in Figure 7 to represent a test or validation of a valency-based scaffold/client model, or a result stemming from the previous data in the manuscript. Rather, we intended it simply as an empirical correlation that could potentially be useful in making predictions about other condensates where quantitative imaging data are not readily obtainable".This does not seem to match the article's title, which promises "compositional principles". Their offer to image two components are the same time will not add much beyond confirming what they have already shown?

We have removed the quantitative correlations between valency and physical features of P bodies (original Figure 7), which was problematic for a variety of reasons. Our two-color experiments have provided important tests of potential models for condensate formation. Further, as stated above, we have extensively revised our model.

We believe that the manuscript provides several principles regarding the composition of P bodies. These are described in the revised Discussion section of the paper, which is completely rewritten. These include: A) the concept (supported by evidence) that biological condensates that appear quite complex by qualitative analyses of their components, may in fact have a much simpler primary complexity and organization when considered quantitatively; B) the idea that differences between highly concentrated and weakly concentrated P-body proteins are probably closely related to their intermolecular connectivity patterns within P bodies; C) the idea that interactions that produce condensates are distributed across their highly-valent components, these act with differing degrees of cooperativity to promote formation of the larger assembly; and D) the prediction that thermodynamics of forming a condensate and the composition of the resulting structure should be coupled, based on the connectivity patterns in the interaction network; i.e. deletion of a highly connected molecule should affect both the concentrations of other factors needed to form the condensate as well as the relative concentrations of the components in the condensate.

Finally, the statement that "our data show that only a handful of the many proteins in P bodies are highly concentrated in the structure. This information is key to conceptualizing the constitution of condensates. It speaks against most prevailing models of condensate composition, which cast the compartments as intractably complex, with hundreds of components, all implicitly weighted equally in terms of functional significance. Our data suggest the compartments are much simpler and provide the first routes to reliable biomimetic reconstitution".Biomimetic reconstitution of RNA granules has already begun, and it is already been shown that few components can be sufficient to mimic condensate structure (Feric et al., 2016, Putnam et al., 2019). The statement also seems to imply that knowledge of the concentration of each factor will help identify such key components, a concept that remains untested."

While the work cited in Feric and Putnam is elegant, both utilize only two components in nucleoli and P granules, respectively. Thus, the degree to which they capture the properties of the cellular structures is unknown. Moreover, to our knowledge, it has not been established that the components used in these reconstitutions are, quantitatively, the dominant components of the cellular structures, or whether other components are equally concentrated. Thus, again, the degree to which these reconstituted structures reproduce the biological structures is uncertain. In our view, a critical advance in biochemical reconstitution of condensates that closely resemble cellular condensates is knowledge of the dominant (i.e. most concentrated) components, and their dynamic properties in vivo. With this information, one can combine the appropriate molecules in vitro at their total cellular concentrations and learn whether they form condensates of appropriate component concentrations (both relative and absolute) and dynamics. If they meet these criteria for multiple components, then one can have some confidence that the biochemistry truly is biomimetic. Without this prior information, one is making an educated guess about which proteins to combine biochemically and how well the reconstituted structure mimics the cellular structure.

Summary from a reviewer: This manuscript reports on a quantitative analysis of P body composition in yeast. The authors characterize the concentration, enrichment relative to cytoplasmic pool, and dynamics of 31 reported P body proteins tagged with GFP. They identify 19 that were sufficiently concentrated in P bodies under the conditions tested and divide them into two classes: a "core group" comprising 7 proteins with high enrichment and low dynamics, and the remainder 12 proteins with low enrichment and high dynamics. The authors argue that these observations support a scaffold/client model for condensate assembly where the number of interaction partners for a given protein (valency) can be used to predict its enrichment in condensates/P bodies.

We have extensively revised the text, and no longer present the work as testing a scaffold/client model for condensate assembly. In fact, based on our data we now present a new, more nuanced view of the scaffold/client nomenclature that better captures the behaviors of natural condensates and will be more useful to the field. We also have removed the analyses describing quantitative relationships between P body enrichment and valency of interactions. We do argue, however, that there is a meaningful qualitative relationship between connectivity in the P body interaction network and molecular behaviors, with a variety of caveats that are now detailed in the Discussion. We say more about each of these issues below.

The main concerns of the reviewers can be summarized as follows:1) The novelty of the work is not emphasized. Clearly it is more comprehensive than previous work of these authors, but it needs some more perspective and additional testing.

We have extensively revised the Discussion to highlight the new principles that we feel derive from our data. First, our work is the first comprehensive, quantitative analysis of the composition and dynamics of any natural biomolecular condensate. Second, this quantitation has revealed that despite the large number of molecules present in P bodies, only a small number (7) are highly concentrated there with large partition coefficients. This is a large reduction in complexity when considering the formation, regulation and function of P bodies. Third, the quantitation has revealed that only a small fraction of most proteins are present in P bodies, which has important implications for functions of the condensates. Fourth, most (6 of 7) of the highly concentrated proteins/complexes are highly connected in the P body interaction network, and none of the weakly concentrated proteins are highly connected. These correlations suggest that connectivity plays an important role in governing the composition of the structures. It is important to note, though, that other factors, including binding affinity and active processes also contribute to molecular behaviors, and we now explain such complexities in our discussion. Finally, based on our data and other data in the literature, we have revised the notion of scaffolds and clients in biomolecular condensates. We argue that these terms should not be used to classify molecules into binary groups, as we had done before, but rather as qualitative descriptors of the degree to which a molecule contributes to formation and composition of a condensate. Thus, a molecule should be described as more scaffold-like or more client-like depending on whether has greater or lesser effects, respectively. This use of the terms better reflects experimental data, while still capturing the idea that some molecules contribute more than others to the formation and composition of a condensate.

2) The authors need to consider other models for assembly than just client/scaffold. The discrepancies between the data and their model are not resolved. There are several experiments detailed that could do more to enlighten the murky conclusions (see below). For instance, rather than tagging with a single fluorophore, can they estimate composition and variation better with a two fluorophore pairwise analysis.

We now provide a much more nuanced view of P body assembly, emphasizing the role of highly-connected molecules (both proteins and RNA) in producing the condensate, but also discussing the importance of binding affinity and other parameters. As requested by the reviewers we have also performed 2-color imaging to examine correlations between the enrichment between different molecular pairs (new Figure 3). These data revealed significant correlations (Pearson’s R of 0.6-0.7) between Dcp2 enrichment and that of Edc3, Pat1 and Xrn1. The Dcp2-Xrn1 correlation is particularly interesting, as the two proteins are not known to bind each other directly, suggesting correlations mediated by indirect connectivity in the condensate (likely through RNA or other core P-body components). The correlations are not so strong, though, to suggest that a stoichiometrically-defined complex is at the heart of P body formation (a model suggested by one of the reviewers). Moreover, as we argue below, existing data speak against a model where RNA is the only scaffold-like molecule in P bodies (a second model suggested in the review), as a number of proteins have also been shown genetically to play important roles in generating the condensates. These issues – cooperativity in recruitment, the possibility of a stoichiometric complex, the roles of RNA-RNA and RNA-protein interactions, and other factors – are now considered in some detail in the Discussion section of the manuscript. We hope this more nuanced view of condensate formation will appeal to the reviewers.

3) Can they provide any additional data or insight to shed light on the discrepancy that they point out between their model and the results for DCP2?

We have addressed this in two ways. First, we have made additional Dcp2 mutants to understand better how its binding to Edc3 controls it enrichment in P bodies (new Figure 6). These lead to an important conclusion that when affinity between two condensate proteins is low, increasing affinity can increase enrichment, but when affinity is already high, increasing it further does not increase enrichment. Thus, tuning of enrichment likely occurs through alterations in the low/modest-affinity regime. Additionally, as described above, the new discussion provides a more nuanced view of recruitment, as deriving from a combination of connectivity, affinity and active processes. When considered together, these explain the behaviors of our Dcp2 mutants. Essentially, not all regions of Dcp2 contribute equally to enrichment; deleting a high-affinity binding element (to either RNA or Edc3) has a much more pronounced effect on enrichment than deleting a low-affinity element.

Reviewer #1:[…] To further improve the novelty, we suggest additional questions that could be addressed:1) A major advance of this study is the precise concentration measurement of GFP-fusion proteins, which is done by comparing the cellular GFP intensity with standard curves. All conclusions depend on these standards and background subtraction. Even though the Rosen lab previously published a paper using a similar approach (Banani et al., 2016), the details of these calibrations and background subtraction should be reported in the supplemental materials.

We have added a detailed description of the image acquisition (shown in Figure 1—figure supplement 2) and analysis procedures in the Materials and methods section.

2) To estimate the protein concentrations in P-bodies, the authors made a fundamental assumption: that all P-bodies have the same and constant composition. However, in Figure 1, the authors showed that the core components of P-bodies, which are also the most abundant proteins, have strong (4-5-fold) variability in concentration and partitioning coefficient. It is unclear how this variability affects the relative quantifications reported in this manuscript (Figure 3). A control experiment that could address this question would be to tag several P-bodies components with spectrally distinct fluorescent proteins (2 or 3 at the time) and verify that the expected relative ratios are recapitulated.

We note that in measuring the absolute concentrations of the various species, we have not made any assumptions about whether P bodies have constant or variable compositions. Assuming that GFP-tagging does not alter the concentrations of a given protein (which our data in Figure 1—figure supplement 3 support) our data capture the range of concentration values sampled by each component across a population of cells. These data show that compositions have substantial variability, as the reviewer notes.

In interpreting the data, the question of relative variability becomes important. Here, we thank the reviewer for suggesting multi-color experiments. In a new Figure 3 we have now used 2-color imaging to simultaneously quantify the concentrations of three pairs of proteins in individual cells, Dcp2/Edc3, Dcp2/Pat1 and Dcp2/Xrn1. We find that in all cases, the P body concentrations of the pairs are significantly correlated (Pearson’s R of 0.6-0.7). Thus, the concentrations of these components fluctuate together (albeit with still some remaining variability, as the R values are not 1). Given the high connectivity of interactions between P body proteins, we believe that many components likely show similar correlations, but a more extensive analysis of this point is beyond the scope of this already lengthy study.

Whether there could be multiple types of P bodies, with different compositional profiles, in a cell population is a difficult question to address. We do not see strong evidence for multimodal distributions in the concentration profiles, which speaks against this possibility. But again, addressing this possibility comprehensively would require vastly more data to assess whether the concentration distributions could be fit better to single, or multiple populations.

3) Along the same line, it is unclear whether the GFP-tagged proteins, which were used to quantify the P-body components in the dcp1∆ background, were expressed from the endogenous locus (Table 1: YRP1936, yRP2254, yRP2237, yRP2246, yRP2230, yRP1840, yRP1736, yRP2269, yRP1844, yRP1916, yRP1842). The authors should specify this since it would have significant consequences on the quantifications if there were a mixed population of tagged and untagged proteins.

All GFP-tagged and mCherry tagged proteins used to generate Figures 1-4 were expressed from their endogenous locus. This is stated in the main text and in the Materials and methods.

4) In Figure 2, The authors found a negative correlation between the fraction recovery rate and P-body concentration. However, the molecular mechanism behind this correlation is not discussed. The authors should expand the discussion and link it to the interaction valency.

As suggested by the reviewer, in the Discussion we now link connectivity in the network (the number of different molecular types a given protein interacts with, which is related to interaction valency, the number of different molecules a given molecule interacts with) to the FRAP behavior. We state “With the exception of Xrn1 (see below), the core proteins all have high valency of interaction (number of interacting molecules) and high connectivity to other P body proteins and RNA (≥ 4 directly interacting molecules). […] These features should generally afford lower P body concentrations and more rapid and complete exchange.”

5) In Figures 4 and 5, the authors measured the partition coefficients of several Dcp2 mutants. It would be important to demonstrate that these mutations do not significantly affect the protein stability and expression by a western blot of these mutants.

In Figure 5—figure supplement 1B and 1C, and Figure 7—figure supplement 1A and 1B, we now show western blots for nearly all Dcp2 mutants examined by microscopy. These data show that proteins compared in the same figure express at approximately the same level. Mutants not examined by western blotting were examined by fluorescence imaging to compare total expression level in the cells analyzed (Figure 6—figure supplement 1). The Dcp2 mutants compared in Figure 6 do express at somewhat different levels, such that Dcp2C∆5H, which partitions weakly into P bodies, expresses ~30% higher than Dcp2C∆5H-H1, which partitions much more strongly (Figure 6 and Figure 6—figure supplement 1). However, the direction of this difference actually strengthens our conclusion that the lower partitioning of Dcp2C∆5H was not because of reduced expression.

6) In Figure 5, the authors suggest that the decapping activity of Dcp2 affects the partition coefficient and recovery rate. However, as the authors mentioned in the text, it is not clear whether this is due to an increase in cytoplasmic mRNA or a direct result from Dcp2 catalytic activity. This issue may be resolved by expressing Dcp2 variants in wild type cells and inducing P-bodies with starvation.

We thank the reviewer for this suggestion. We have now performed this experiment, and the results are summarized in a new Figure 7. In brief, under starvation conditions the dynamics and distribution of number of P bodies are the same for the two mutants, suggesting that these parameters are mainly responding to mRNA levels when catalysis is impaired. However, the total fraction of material in P bodies is still higher for the catalytic mutant during starvation. Thus, this parameter of P bodies appears to be responding to the loss of catalytic activity. We conclude from these data that both the increase in mRNA levels and loss of catalytic activity account for the behavior of the WD mutant.

7) In Figure 7, the authors found a correlation between P-body partitioning and protein interaction valency. This finding would be more convincing if the authors could demonstrate that when the interaction valency of a protein increases, this protein will become more concentrated in P-bodies. For example, this can be done by fusing a client protein with a domain of a scaffold protein, like Dcp2C ∆5H. The fusion protein should have more interaction valencies, and the fusion protein would be expected to be more concentrated in the P-bodies. This experiment will also enhance the conclusion made in Figure 6.

We have removed Figure 7 from the paper, and no longer discuss quantitative relationships between valency and partitioning. The problem with invoking valency alone is that it ignores affinity (an issue that we mentioned in the previous text, but did not emphasize strongly enough), and affinity plays an important role in the partitioning of Dcp2. In the revised manuscript, we have addressed this in two ways. First, we now explicitly state that data here, and in the literature, indicate that RNA binding by the N-terminal domain and Edc3 binding by HLM1 both occur with high affinity, and contribute strongly to Dcp2 partitioning, while the C-terminal HLM elements are lower affinity and contribute less (Figure 5). Second, we have done experiments analogous to those suggested by the reviewer. We added the high-affinity Edc3-binding motif, HLM1, to both Dcp2 ∆5H, which partitions weakly, and to Dcp2 wild type, which partitions strongly. We find that addition of HLM1 only increases the partitioning of the former protein (Figure 6D). We conclude from these data that when affinity for Edc3 is low (Dcp2 ∆5H), adding an additional high affinity binding site can increase partitioning of Dcp2. But when affinity is already high (Dcp2 wild type), increasing it further has no effect. This result is likely general in considering recruitment of proteins into condensates.

Reviewer #2:[…]This model was derived by the Rosen group in a previous study that examined the in vitro phase separation properties of an artificial set of proteins with varying numbers of low-affinity binding motifs (valency). The model predicts that "scaffolds" (proteins essential for condensate assembly) exhibit high enrichment and high valency, whereas "clients" exhibit low enrichment and low valency. In accordance with the model, the authors note a rough correlation between valency and enrichment among the 19 P body proteins examined (but see below). To test the model directly, they deleted binding motifs in one predicted "high-valency" P body protein (Dcp2). Surprisingly, they found no strong correlation between valency and enrichment (Many of the Dcp2 mutants do not lie on the correlations in Figure 7. It is not clear to us how to resolve this discrepancy). They also find that some known clients (proteins not essential for P body assembly) exhibit strong enrichment, contrary to the model. Despite these discrepancies, the authors continue to argue that their findings support the scaffold/client model.

We have extensively revised the text and our analyses of our data to address these critiques. Most significantly, we have removed Figure 7 and eliminated all discussion related to a quantitative correlation between valency and P body concentration. As the reviewer states, valency alone cannot account for our data on Dcp2 mutants. We now present a much more nuanced view of how concentration and dynamic properties of P body components appear to be determined. We believe that the connectivity of interactions (the number of molecule types contacted by a particular molecule) plays an important role, as most (6 of 7) of the highly concentrated proteins are also highly connected (≥4 interaction partners), and all of the weakly concentrated proteins have low connectivity (≤2 interaction partners). Further, most (the same 6 of 7) of the highly connected molecules have also been shown to contribute to P body assembly, and none of the low connectivity molecules have been shown to do so. Nevertheless, connectivity is not the only factor, and binding affinity and active processes can also be very important. In this light, we now discuss how interactions of the Dcp2 N-terminal domain with RNA and of the HLM1 element with Edc3 are of higher affinity than other Dcp2 interactions, and thus play more important roles in determining the concentration of the protein in P bodies (explaining the deletion data). Similarly, Xrn1 binds RNA with high affinity. Thus, even though its connectivity is low, it is strongly concentrated in P bodies and shows moderate dynamics. In terms of active processes, ATP hydrolysis by Dcp2 also contributes to the dynamics of the protein. Thus, connectivity is important, but not the whole story in defining the behavior of condensate molecules. These issues are discussed in the new Discussion.

Through these considerations we have come to a different view of how the notions of scaffold and client should be used. Rather than classifying molecules as either scaffold or client, we now feel that these terms should be used as descriptors. A molecule is more scaffold-like if it plays a greater role in P body assembly, and is more client-like if it plays a lesser role. In this way, we can account for the fact that different molecules have different effects upon deletion. E.g. RNA appears to be the most scaffold-like component of P bodies, as its elimination by RNAse destroys the condensates (consistent with the fact that all 19 P body components quantified here bind RNA), while Edc3 and Pat1 are less scaffold-like, since their deletion strains retain some P bodies, albeit fewer and smaller than wild type strains. In contrast, all of the 12 non-core proteins we studied are client-like, since none has been shown to significantly decrease P body assembly when deleted. We note that connectivity (and probably network centrality) does appear to play a role in determining whether a molecule is more scaffold-like or more client-like, as the most connected molecules are all scaffold-like (and more central in the P body interaction network) and molecules with few connections are all client-like (and more peripheral in the network). These issues are discussed in the new Discussion in a new section titled “General principles of scaffolds and clients in natural condensates.”

It is very surprising that the authors do not consider other models that might fit their data better. One possibility is that RNA functions as the true scaffold for P bodies and that proteins are recruited to P bodies by virtue of their affinity for RNA or for other proteins that binds RNA. In this regard, it is interesting that components of multiprotein complexes are found in roughly stoichiometric amounts in P bodies, suggesting that complexes are preserved in the condensates. In this alternative model, high affinity for RNA by one protein in the complex would be sufficient to enrich the entire complex, even though none of the other subunits in the complex bind to other P body components.

As we understand this comment, the reviewer is suggesting that P-bodies form through interactions between mRNAs and then those mRNAs recruit the P-body core proteins by high affinity RNA binding. However, several observations in the literature argue against this model. For example, as we point out in the manuscript, P body formation clearly requires protein-protein interactions since Dcp2, Edc3, Dhh1, Lsm1-7 complex and Pat1 have all been genetically shown to promote P-body formation (e.g. Decker et al., 2007; Sheth and Parker, 2006; Hondele et al., 2019; Rao and Parker, 2017). Moreover, the mechanisms by which proteins promote P-body formation can be directly linked to specific protein-protein interactions, such as the dimerization of Edc3 (Ling et al., 2008). In addition, some proteins require other P-body components for their recuritment to P-bodies. For example, Dcp1 requires Dcp2 to be recruited to P-bodies and the Lsm1-7 complex requires Pat1 (Teixeira and Parker, 2007). Thus, P-body formation, and recruitment of core P-body components to P-bodies, requires protein interactions.

We agree with the reviewer that RNA contributes to P-body formation. Clear evidence suggests that a pool of untranslating mRNAs is required for P-body formation (Teixeira et al., 2005). One clear role for RNA is to provide binding sites for interacting proteins thereby allowing P-body assembly. In this role, we agree that the RNA is functioning in a scaffold-like manner. Whether RNA also contributes to P-body formation through intermolecular RNA-RNA interactions remains to be established. Regardless, a model where RNA is the sole scaffold, which simply binds the protein components of P bodies is unlikely. To clarify this issue in the manuscript, we have added this argument in the new Discussion.

Moreover, as described above, we now present a much more detailed and nuanced view of P body assembly that does not hinge on a simple, binary classification of molecules as either scaffolds or clients. The revised Discussion also addresses the second model proposed here by the reviewer, that the highly concentrated proteins form a stoichiometric complex, which then assembles on RNA to produce the larger structure. A stoichiometric complex would show extremely high cooperativity in recruitment into P bodies, since all elements would enter or exit together. While new Figure 3 does show correlations between the P body concentrations of Dcp2 with Edc3, Pat1 and Xrn1, the correlations are not sufficiently high (Pearson’s R values of 0.6-0.7) to suggest a stoichiometrically defined assembly, even though the proteins have roughly equal average concentrations in the condensate. Further, the P body concentrations of Pat1 and the Lsm1-7 complex drop about two fold in wild type strains under glucose starvation conditions relative to the *dcp1Δ* strains, again speaking against a discrete assembly. These arguments are presented in a new paragraph in the Discussion.

The authors also claim that their analyses suggest that a relatively small number of core proteins (7) make up most of the protein content in P bodies. However, since other P body components may have yet to be discovered, this statement can only by an approximation at this time. Furthermore, even if true, the significance of this hypothesis is unclear – proteins present in low concentrations in P bodies may still have important roles.

We have removed this claim from the text.

In conclusion, while this survey documents concentrations and dynamics for a significant number of P body proteins, it does not appear to provide significant insights into the "rules" that regulate P body composition. The model put forth by the authors is not consistent with the data and other more plausible models are not considered.

As described in our response to the first summary comment at the top of this review, we feel that our work has revealed a number of new and important principles regarding the formation and composition of P bodies. The new text presents a much more nuanced description of P body assembly that is now consistent with all of our data, and considers a variety of different mechanistic issues and possibilities.

Reviewer #3:[…]- Limits to novelty. It was already known that these 9 proteins are the major proteins of the P body that also play some role in P body function. These earlier studies had also already shown that several of these proteins contain multiple interaction domains through which P body components interact.

As described in response to Summary point 1 at the top of this letter, we feel that our work is novel in many respects, including being the first comprehensive, quantitative analysis of any biomolecular condensate. Through this quantitation we revealed two distinct classes of molecules based on concentration in the structures, something that could not have been inferred from previous work, and also the important idea that for nearly all components, only a small fraction of the molecules are present in P bodies. Analysis of these data in light of known molecular interactions also revealed correlations between the physical behaviors of P body components and their molecular connectivity, binding affinity and ATP hydrolysis activity. Again, these could not have been inferred from previous data, as they depend inherently on our quantitative analyses. If the reviewer can point us to relevant literature documenting that the core P-body proteins we identify as being highly enriched have higher partition coefficients than other identified P-body components, we would be happy to revise our manuscript accordingly.

- Limits to quantitative analysis. Analysis with confocal has clear limits for spatial resolution and the quantitative measurements are based on fluorescence intensity, not single molecule measurements. Since all components are tagged with the same fluorophore (GFP) no pairwise or higher order analysis could be used to determine variation between granules, protein occupancy, correlation between proteins, to test, as example, whether all components are in a granule at same time, and with similar concentration.

As detailed above, we have now performed 2-color imaging of several pairs of P body proteins, and find substantial cooperativity between their enrichment. This has led us to a more detailed view of P body assembly, which is described in a virtually completely rewritten Discussion section of the manuscript.

- Unclear relevance of DCP2 analysis. The analysis of DCP2 seems not add much to the conclusions of the paper. Rather than showing that this protein is multivalent as has been shown by domain dissection of other “scaffold” proteins in the past, it would have been informative if the authors had provided whether the “client” proteins have indeed fewer interaction domains.

Our analyses of Dcp2 have dissected the mechanisms by which it is enriched within P bodies. Although the notion of multivalency has been demonstrated for other proteins as well, this dissection provides useful information about Dcp2. Further, in studies of Dcp2 we have found two previously unrecognized properties that are likely relevant to components of other condensates. First, we find that two fragments of the protein that partition only weakly into P bodies individually, partition strongly and cooperatively when fused together. This occurs even though the fragments bind to different P body components, and thus is not the same as simple avidity. This result was far from obvious to us when we initiated the work. Second, in new data presented in new Figure 6, we find that adding a high affinity Edc3-binding element to a Dcp2 protein only increases partitioning into P bodies when the Dcp2 has low affinity for Edc3. When affinity is already high, adding more binding elements has no effect. Thus, partitioning of a condensate component can be saturated, likely due to limiting concentrations of binding partners. Again, this was not an obvious result when we began these experiments.

Regarding proteins with client-like behaviors (see response to Summary point 1 for our new way of using this term), as detailed above, we show in Table 3 and Figure 2—figure supplement 3 that all molecules that are weakly concentrated in P bodies have low connectivity to other P body proteins and are not known to contribute strongly to P body assembly. So indeed, client-like behavior is correlated with limited interactions with other P body components. The correlation is not universally true in the opposite direction. That is, there is small number of proteins that have only few connections to other P body components but yet are highly concentrated in the condensate (Xrn1, Dcp1 and Pby1). These proteins are known to have high affinity for another highly concentrated P body component, which recruits them. These issues are now covered in detail in our revised Discussion.

[Editors’ note: further revisions were suggested prior to acceptance, as described below.]

Revisions:The reviewers felt the manuscript and work makes a contribution to the field, but some revisions are necessary to take into account caveats in the interpretation of the data. For instance, reviewer 2 feels that the term "cooperativity" is not appropriately supported by the data and that mRNA is not considered as a component of the P bodies.

We have changed our description of the data in Figure 5F, the source of this claim, to “synergistic”, which states that the effect of combining two components together results in a larger effect than the sum of the components individually. In the case of Dcp2, the Dcp2 300ΔH1 construct has a PC of 2.5, and the Dcp2C Δ5H construct has PC of 3.5, while the of Dcp2ΔH1 Δ5H construct, which combines Dcp2 300ΔH1 and Dcp2C Δ5H has PC of 31, which is > 2.5*3.5. Thus, the system meets the definition of synergy. We now note that this effect may result from cooperative binding to molecules in the P body.

The manuscript mentions RNA as an important P body component. We feel that this is sufficient, especially given that the paper is focused on an inventory of P body proteins, not RNAs.

Reviewer 3 feels that the functional properties are not sufficiently addressed, particularly in view of specificity (why are some mRNAs and RBPs excluded?).

Our study does not attempt to address the function of P bodies. This is a complicated issue that would require much more and different data than we can realistically put in the manuscript. Our work does address the specificity of protein recruitment in a significant way through the idea of synergistic recruitment of Dcp2 to P bodies. That is, proteins that are resident often contain multiple weak recruitment elements, and by implication proteins that contain few such elements are recruited poorly if at all. RNA recruitment may follow similar principles, and we now mention this idea explicitly.

Reviewer #2:In this largely revised manuscript, the authors report on a comprehensive quantification of component proteins of a biological condensate, the Dcp1 depleted P bodies of yeast. Through this they identified seven “core” P body proteins that are partitioned substantially, have a high concentration and slow mobility when enriched in P bodies (Figure 1 and 2). Concentration of these seven proteins in P bodies is positively correlated, consistent with previous data showing that these proteins interact biochemically (Figure 3). The reported enrichment parameters seem similar when Dcp1 depleted P bodies are compared with wild type P bodies under glucose starvation, however Dcp1 and its binding partner Pby1 are missing and RNA degradation is defective in the P body model used throughout the analysis. Thus, while at the level of their biophysical analysis the Dcp1 depleted P-bodies seem similar to P bodies under physiological conditions, functionally these “P bodies” are distinct. This reduces the relevance of this study.

The reviewer is correct that we have studied P bodies under 2 different conditions, only one of which is wild type. We note that this is not wholly different, however, from most studies in the field. For example, most analyses of stress granules induce them with arsenic, a toxin that is only rarely encountered in natural biology. Many other studies involve overexpression of individual components, again a non-physiologic situation. There is nevertheless substantial value in such analyses in understanding the principles by which condensates are assembled, and we feel that our study is in a similar vein. While Dcp1-/- P bodies and glucose starved wild type P bodies may have different functions, what is striking is that their compositions are closely related. Moreover the key patterns of composition are retained – a small number of highly concentrated proteins and a larger number of weakly concentrated proteins – which is an important feature of these cellular structures that has not to our knowledge been examined elsewhere (even the 3 recently published Cell papers on stress granules only reported partition coefficients, which lacks information relative to the absolute concentrations reported here).

In Figure 4 the authors observe that core protein concentrations in P bodies are largely independent of stress conditions, however the authors only considered the protein components, while the regulated entity of stress P bodies are mRNAs and not proteins.

We state that our data do not address changes in mRNA content during starvation: “… nor do they speak to sequestration/storage of RNA…”. We also note that the main point of Figure 4 is to show that for most proteins, P bodies do not substantially deplete the amount of molecules from the cytoplasm, not to compare Dcp1-/- and glucose starved wild type P bodies. If the reviewer is interested, recent results from mammalian cells suggest the composition of P-body mRNAs appears to be largely regulated by their translation rate, perhaps with an additional input from the poly(A) tail length (Matheny et al., 2019).

Next the authors dissect DCP2 P body localization in a Dcp1, Dcp2 double null mutant, which still makes some P body (although Figure S1A only shows the presence of small “P body like” structures but no quantitative comparison to control P bodies, in wild type under glucose starvation and Dcp1 depleted).

We are not entirely sure what the reviewer is asking for here, as we have quantified the PC and dynamics for the various re-expressed Dcp2 mutants. We have now quantified the size distribution of the Edc3 puncta in the Dcp1, Dcp2 double null mutant shown in Figure S5A (Figure S1A is unrelated to these experiments; we assume a typo from the reviewer), and report that in Figure 5—figure supplement 1A.

In deletions studies the authors find that various domains of Dcp2 contribute to P body partitioning and show some redundancy between the H1 domain and other helical domains in the CTP. The authors conclude that elements in Dcp2 act “cooperatively”. However, their data show additivity or at most synergy but not cooperativity. In order to use the term cooperativity, the authors need to demonstrate that the initial binding event increases the binding affinity of subsequent binding events.

As stated above we have changed our description of the data in Figure 5F, the source of this claim, to “synergistic”, which states that the effect of combining two components together results in a larger effect than the sum of the components individually. In the case of Dcp2, the Dcp2 300ΔH1 construct has a PC of 2.5, and the Dcp2C Δ5H construct has PC of 3.5, while the of Dcp2ΔH1 Δ5H construct, which combines Dcp2 300ΔH1 and Dcp2C Δ5H has PC of 31, which is > 2.5*3.5. Thus, the system meets the definition of synergy. We now note that this effect may result from cooperative binding to molecules in the P body.

The authors use structure-function study of DCP2 to argue that connectivity, affinity and catalytic activity are three major factors for DCP2 P body recruitment and P body size and number. However, more specific experiments would need to be performed to strengthen this argument or to generalize the finding to other P body components or other types of RNP bodies: 1) doing domain/binding site deletion study on Edc3 or other core proteins; 2) adding valency or high affinity binding site to a client-like protein to turn it into a scaffold (sufficiency). 3) Testing the model with stress granule scaffold proteins.

We agree that our arguments will be strengthened by additional experiments, and these are planned for additional projects and publications in the future.

Overall, this study is technically well done but rather limited in its take home message or novelty. A true functional readout is missing as the experimental set up does not relate to P body function (such as the level of RNA in different granules and decapping activity). The manuscript could easily be condensed into four figures, each making one point for the characterization of a particular type of condensate: Figure 1 and 2: enrichment of certain factors in P bodies, Figures 3 and 4 relationship of concentration between components and under differing conditions, and Figures 5 and 6 structural analysis of one component, suggesting partial redundancy between the CTD helices and Figure 7 with studies addressing RNA binding.

We agree that we have not attempted to address P body function in this work. We believe that such analyses would require a publication in their own right. Given the lack of figure constraints in *eLife*, we have not compressed the data into a smaller number of figures, as we fear this would make the paper harder to read and understand (e.g. Figure 4 is not meant to compare different conditions, but rather to make the point that most molecules are not present in P bodies for most species. Hence its title “P bodies do not strongly sequester their resident proteins.”

Reviewer #3:In this revised manuscript, Xing, et al. analyze the enrichment and dynamic properties of ~20 P body proteins in yeast. The descriptive part of the study is reasonably comprehensive (31 proteins included in the initial survey, 19 whose abundance was quantified) and suggest the existence of two groups: proteins strongly enriched in P bodies – several of which are also required for P body assembly – and proteins that are less enriched – fewer of which affect P body assembly. From this the authors suggest that P bodies are "biochemically simpler than suggested by proteomics". This is at best a conjecture and at worst a gross oversimplification since abundance is not necessarily predictive of function – especially for enzymes.

For clarity, we have changed the conclusion to “compositionally simpler than suggested by proteomics”. This should address the reviewer’s principal concern. We note, however, that our data in Figure 4 speak against the reviewer’s assertion that abundance does not predict function. That figure shows that for nearly all proteins enriched in P bodies, the large majority of the molecular species are *not* in the body but rather are present in the cytoplasm. For all but the top 6 molecules, less than 20% is in the body, even when conservatively correcting for non-observable sub-diffraction P bodies (without this correction the value drops to 8%). Thus, it is hard to claim that the amount in the P body makes a large difference to total activity of that species in the cell. There are ways around this problem involving, for example a huge increase in specific activity (activity per molecule) within the P body versus the cytoplasm, or co-concentration of multiple molecules within a cascade. But these require additional assumptions, and the first order view in our opinion is that abundance does play an important role in framing the biochemical functions that arise from condensates, and since only a small number of molecules are highly concentrated, the compartments are, in fact, likely to be biochemically simpler than one might imagine from a proteomics study that does not determine concentrations and thus essentially weights all components equally.

In the second part of the study, they authors examine the domains in DCP2 required for P body localization and identify multiple domains that appear to function partially redundantly. They conclude that a collection of synergistic interactions drive P body assembly, a conclusion reached by several other studies (reviewed in Lu Na and Slavoff, 2018) including in vitro analyses (Shutz et al., 2017).Overall, we are left with a study that will be a useful reference for future P body studies, but that does not deliver on the promise of "principles of composition and specificity". The question of specificity, in particular (why don't all RNA-binding proteins localize to P bodies?) is not examined directly. The authors only perform structure-function studies on one P body protein and never address what excludes other RNA-binding proteins from P bodies.

As stated above, we feel that our work does address the specificity of protein recruitment in a significant way through the idea of synergistic recruitment of Dcp2 to P bodies. That is, proteins that are resident often contain multiple weak recruitment elements, and by implication proteins that contain few such elements are recruited poorly if at all. RNA recruitment may follow similar principles, and we now mention this idea explicitly.

Our work also suggests to us that other RNA binding proteins do accumulate in P-bodies, but their level simply reflects the number of available binding sites in P-body mRNAs. Specifically, while the MS2 RNA binding protein does not typically accumulate in P-bodies, the addition of multiple binding sites for this protein in an mRNA resident in P-bodies can lead to the accumulation of MS2 in P-bodies. Second, we do observe other RNA binding proteins in P-bodies, they just partition less effectively into P-bodies than the core components (Figure 1).

We note that exclusion of molecules from a condensate (PC < 1) will be very hard to observe and meaningfully quantify in cells, given the small size of the structures. Further, it is more likely that most molecules will simply not be recruited (PC ~1) than be overtly excluded; we are not aware of molecules that are excluded from P bodies. In general, the physical mechanisms that do afford exclusion from condensates in vivo are not understood at all, and are well beyond the focus of the present study.

We also note that to our knowledge, this is the first time that synergy has been quantitatively demonstrated for recruitment of a protein into a condensate in vivo. The Sprangers work mentioned above is interesting but is purely in vitro. In cells, to claim synergy one would need to delete the protein and then quantify the partition coefficients for different fragments when reintroduced individually and together, as we have done. We are not aware of that level of quantitative dissection having been done elsewhere.

Additional comments:1) The authors are not consistent in the description of the CTD. They suggest first that "N-terminal domain, HLM1, and C-terminal domain are all required for efficient partitioning and maintaining the characteristic slow dynamics of Dcp2". However, later they say that "deleting the CTD has no effect on PC of full length Dcp2". The data presented suggest that the C-terminal domain is not required (300-end) but can compensate when HLM1 is absent.

We thank the reviewer for catching this discrepancy. We have changed the first statement to “ Since the N-terminal domain and HLM1 are required for efficient partitioning and maintaining the characteristic slow dynamics of Dcp2, and the C-terminal domain can compensate when HLM1 is lacking, we conclude that elements controlling partitioning and dynamics are distributed across the protein.”

2) The intact C-terminal domain is never tested; instead the authors used a CTD with 5 of the HLM domains mutated. While the authors imply that this version of the CTD has weaker association with P bodies, no data is shown, and it is unclear from the text why this version of the CTD was used.

We had these data, and now show them in Figure 5—figure supplement 1C.

3) For the data in Figure 7, the authors use DCP2 constructs with reduced binding to Edc3 to investigate the role of RNA binding and decapping activity in P body association. However, it is not clear why RNA binding was studied using the Dcp2 300 mutant and decapping was studied in the Dcp2ΔH1 mutant.

These were essentially historical features of the work, relating to when we obtained various constructs. We do not feel that the differences here impact our conclusions.

4) A figure comparing the P body concentration/partition coefficient and exchange rates for the different DCP2 mutants (similar to Figure 2A and B) might be a more intuitive way to compare all the different DCP2 mutants used in this study.

We feel that while such a plot would help in some respects, it would detract in others, and have not included it. The principle detraction is that in Figure 2 we are specifically highlighting correlations between P body concentration and exchange rates. But the points in Figures 5 and 6 is different, and focused on understanding different aspects of Dcp2 recruitment.

5) The DCP2 structure function studies identify domains in DCP2 that promote P body localization. Whether these domains are sufficient to drive a non-P body protein into P bodies is not addressed. Without such "sufficiency" experiments, whether the results for DCP2 are generalizable for other proteins remains unclear. For example, they suggest that Xrn1 is stably recruited because of its high RNA binding affinity, however, this hypothesis does not address why other high affinity RNA binding proteins are excluded from P bodies.

This comment addresses two points. First, whether domains within Dcp2 that promote P-body localization can act in a dominant manner. Essentially, we have demonstrated this phenomenon with GFP as the foreign protein. A related question, but one beyond the scope of this manuscript is whether the P-body targeting domain of Dcp2 could override features of other RNA binding proteins that either exclude them from P-bodies or target those RNA binding proteins to other condensates. We agree this would be an interesting question and will consider it in future work.

The second point the reviewer raises is that we do not understand why other RNA binding proteins, which can bind RNA with high affinity, do not partition strongly into P-bodies. Two observations suggest to us that other RNA binding proteins do accumulate in P-bodies, but their level simply reflects the number of available binding sites in P-body mRNAs. Specifically, while the MS2 RNA binding protein does not typically accumulate in P-bodies, the addition of multiple binding sites for this protein in an mRNA resident in P-bodies can lead to the accumulation of MS2 in P-bodies. Second, we do observe other RNA binding proteins in P-bodies, they just partition less effectively into P-bodies than the core components (Figure 1).

6) At the end of the Discussion the authors introduce the concept of avidity where multiple interactions result in strong binding. It might be useful to introduce this concept earlier.

We have now introduced the idea of avidity in the Results section on Figure 5.